# Predicting the uniaxial compressive strength and elasticity modulus of sandstones from physical and mechanical properties using statistical analyses and artificial intelligence-based techniques

**Davood Fereidooni**[1]*, **Matloob Hejazifar**[2]

**1** School of Earth Sciences, Damghan University, Damghan, Semnan, Iran, **2** School of Earth Sciences, Damghan University, Damghan, Semnan, Iran

\* d.fereidooni@du.ac.ir

## Abstract

This study develops predictive models for the uniaxial compressive strength (UCS) and elasticity modulus (E) of sandstones by integrating statistical analyses with artificial intelligence (AI) techniques. Comprehensive laboratory tests were performed on 20 sandstone samples from four Iranian formations, measuring key physical (dry unit weight $\gamma_a = 23.34$–$26.10$ kN/m³, porosity $n_e = 2.07$–$10.16\%$) and mechanical properties (UCS = $55.46$–$104.16$ MPa, E = $39.86$–$57.03$ GPa). Statistical analyses revealed strong correlations, with dry unit weight showing the highest Pearson correlation to UCS (R = $0.947$) and E (R = $0.971$), while porosity exhibited significant negative relationships (R = $-0.918$ and $-0.916$ respectively). Among the evaluated AI models, Artificial Neural Networks (ANN) demonstrated superior predictive capability, achieving the highest accuracy for both UCS and E predictions (e.g., UCS-Hs: $R^2 = 0.992$, RMSE = $1.562$; E-Hs: $R^2 = 0.947$). In contrast, Random Forest (RF) and K-Nearest Neighbors (kNN) models provided acceptable but comparatively lower performance, with their best models attaining $R^2$ values of $0.909$ and $0.906$, respectively. Sensitivity analysis identified Schmidt hammer rebound ($H_s$) as the most influential predictor (Univariate Regression score = $3406.678$), followed by point load index (PLI). The PDH formation demonstrated superior mechanical properties (UCS = $97.21$ MPa, E = $54.82$ GPa) linked to its high density ($\gamma_a = 25.52$–$25.99$ kN/m³) and low porosity ($n_e = 2.48$–$4.84\%$), while the HZD formation showed the weakest performance (UCS = $60.25$ MPa) due to high porosity ($n_e = 8.02$–$10.16\%$). These findings provide a robust framework for predicting sandstone mechanical properties using non-destructive methods, offering significant advantages for geotechnical applications where direct testing is impractical.

**Data availability statement:** There are no legal or ethical restrictions for publicly sharing the data generated or analyzed during the current study to support its findings.

**Funding:** This work is based upon research funded by Iran National Science Foundation (INSF) under project No. 4040158.

**Competing interests:** This work is based upon research funded by Iran National Science Foundation (INSF) under project No. 4040158.

## 1. Introduction

Prior to initiating any engineering project, it is crucial to assess the mechanical properties of construction materials, particularly soil and rock. Laboratory testing plays a key role in this evaluation, with standardized procedures such as the Uniaxial Compressive Strength (UCS) test—established by authoritative bodies like the International Society for Rock Mechanics (ISRM) and the American Society for Testing and Materials (ASTM)—being widely employed to determine strength and deformation characteristics of rocks. The UCS and elasticity modulus (E) stand out as fundamental parameters in rock mechanics, serving as an indispensable metrics for geotechnical analysis and design [1,2]. The evaluation of rock mechanical properties has historically relied on two primary approaches: direct and indirect methods. The UCS test serves as a key direct laboratory method for determining uniaxial compressive strength (UCS) and elasticity modulus (E). However, direct methods present significant practical challenges. Obtaining sufficient high-quality core samples proves particularly difficult in jointed, weak, or laminated rock formations [3]. Furthermore, these direct testing methods are often time-intensive and costly to perform [4].

Given the high equipment costs associated with the standard UCS test, indirect testing methods have become a preferred alternative for predicting rock properties. These include p-wave velocity, slake – durability, density, unit weight, Schmidt hammer, point load strength, and Brazilian tensile strength (BTS) tests which offer more economical and practical solutions [5]. To address the challenges of direct UCS determination, Aladejare et al. compiled a comprehensive database of empirical relationships between rock parameters and UCS, consolidating previously scattered and potentially biased literature data [6]. As Ceryan et al. demonstrated, input variables can effectively serve as predictors in statistical models when they show adequate correlations with the target outputs [7,8]. This principle has led numerous researchers to explore relationships between measurable rock properties and UCS/E. Both historical and contemporary studies have focused on developing indirect estimation methods, primarily employing simple regression analyses and fundamental index tests [9–15].

Early research efforts predominantly employed multiple regression analysis to predict the unconfined compressive strength (UCS) and elasticity modulus (E) [16]. However, these methods sometimes yielded limited reliability. Therefore, to overcome the limitations of conventional statistical methods, researchers have increasingly adopted soft computing or artificial intelligence-based techniques. These techniques include many approaches for predicting rock properties. In this regard, artificial neural networks (ANNs) have emerged as a leading approach, demonstrating exceptional capability in addressing rock engineering challenges [17,18]. These models span diverse architectures, including fuzzy inference systems and adaptive neuro-fuzzy systems [19,20], with hybrid methods like Particle Swarm Optimized Back propagation ANNs further enhancing predictive accuracy for rock strength [21–23].

Beyond the previously discussed methods, contemporary research has leveraged sophisticated ensemble and deep learning techniques to enhance the prediction of unconfined compressive strength (UCS) and modulus of elasticity (E). Among these,

boosting algorithms have demonstrated notable efficacy. For instance, Ghorbani and Bameri developed hybrid machine learning models combining meta-heuristic optimization algorithms with XGBoost and RF to predict rock uniaxial compressive strength (UCS) using non-destructive parameters (P-wave velocity, porosity, Leeb hardness, density). Among the six models tested, the Bayesian optimization-random forest model performed best, achieving high accuracy with an $R^2$ of 0.901, NRMSE of 0.192, VAF of 90.1, and MAE of 14.36. This approach provides a more efficient and accurate alternative to traditional laboratory methods and standalone ML techniques for UCS prediction [24]. Liu et al. developed an evaluation framework incorporating extreme gradient boosting (XGBoost), adaptive boosting (AdaBoost), and categorical gradient boosting (CatBoost) to assess sandstone UCS [25]. Hybrid and innovative models have further expanded predictive capabilities. Sun et al. pioneered a novel approach combining X-ray computed tomography with convolutional neural networks (CNNs) for UCS estimation [26], while Cao et al. [27] introduced the XGBoost-Firefly Algorithm (XGBoost-FA) model, integrating optimization and machine learning to predict UCS and E. The model's robustness was validated against support vector machines (SVM) and radial basis function neural networks [28]. Comparative studies of tree-based algorithms have also contributed to methodological advancements. Barzegar et al. evaluated the performance of three standalone models—M5 model trees, multivariate adaptive regression splines (MARS), and random forests (RF)—for predicting UCS in travertine rocks, demonstrating their applicability in geotechnical contexts [29].

Several studies have demonstrated the effectiveness of artificial neural networks (ANN) combined with optimization algorithms for predicting uniaxial compressive strength (UCS) from non-destructive test indicators. Skentou et al. developed three ANN-based models (ANN-PSO, ANN-ICA, and ANN-LM) using P-wave velocity ($V_p$), Schmidt hammer rebound number ($H_s$), and effective porosity ($n_e$) as inputs, finding that the ANN-LM model achieved superior performance ($R^2 = 0.9607$) and subsequently created a Graphical User Interface (GUI) for UCS estimation [30]. Wei et al. developed artificial neural network (ANN) models to predict the uniaxial compressive strength (UCS) of sedimentary rocks using non-destructive input parameters (dry density, wet density, and Brazilian tensile strength). Among the tested models, the M2 ANN model (trained on a 70−30% dataset split) performed best, achieving an $R^2$ of 0.831, RMSE of 0.27672, VAF of 0.92, and a20-index of 0.80 in testing. It outperformed multiple linear regression (MLR) and other ANN configurations, making it the most reliable predictor for UCS in the Thar coalfield, Pakistan. Sensitivity analysis revealed that Brazilian tensile strength (BTS) was the most influential parameter in UCS prediction. This approach offers an efficient alternative to costly and destructive laboratory tests [31]. Complementary research by Armaghani et al. showed that ANN models using just two non-destructive test indicators could outperform existing methods for granite UCS prediction [32]. Similarly, Asteris et al. established correlations between Schmidt hammer types (L and N) and rock strength through soft computing models [33]. Further validation comes from Momeni et al. work in Peninsular Malaysia, where a PSO-optimized ANN model achieved exceptional accuracy ($R^2 = 0.97$) for predicting UCS from basic rock strength parameters [34]. These studies collectively demonstrate that hybrid ANN approaches, particularly when combined with optimization algorithms, provide reliable alternatives to conventional UCS determination methods.

Other intelligence-based methods were frequently used for predicting the uniaxial compressive strength (UCS) and elasticity modulus (E) of rocks from their physical and mechanical properties (e.g., [35–41]). In this regard, Aladejare et al. compiled a comprehensive database of empirical relationships for estimating the UCS of rocks from various rock properties, addressing the challenges of scattered and site-specific models in existing literature. Through statistical analysis, the research found that most regression equations were developed using substantial datasets with moderate to high $R^2$ values. The compiled database provides a valuable resource for selecting appropriate site-specific UCS estimation models, helping to reduce bias and improve accuracy in small to medium-sized mining projects where direct laboratory testing may be impractical [6]. Li et al. demonstrated that AI-optimized models, particularly the LSO-RF (lion swarm optimization-enhanced random forest), outperform traditional empirical equations in predicting rock uniaxial compressive strength (UCS). Using 386 rock samples with four input parameters (load strength, porosity, P-wave velocity, and Schmidt hardness), the AI approach achieved higher accuracy, with porosity (Pn) identified as the most influential factor. The

findings highlight AI as a more reliable and efficient alternative to empirical methods for UCS prediction, offering improved practical applications in civil and mining engineering [42]. Fereidooni and Karimi developed different machine learning models to predict rock brittleness indices (B1-B5) (which are comparable with UCS) using simple laboratory tests (dry unit weight, porosity, P-wave velocity, etc.) instead of complex mechanical tests. Among multiple methods tested, Support Vector Machine (SVM) showed the best performance for predicting B1, B2 and B5, while Regression Tree Ensemble (RTE) was superior for B3 and B4 predictions. The models were developed using 39 diverse Iranian rock samples (igneous, sedimentary, metamorphic) and validated through comprehensive statistical metrics, offering a practical alternative to conventional brittleness assessment methods [43]. Fereidooni et al. developed stacking ensemble models to predict the uniaxial compressive strength (UCS) and elasticity modulus (E) of carbonate rocks using non-destructive test data. The models combined multiple machine learning algorithms (MLP-RF for UCS and SVR-XGBoost for E) and achieved high accuracy, with $R^2$ values of 0.909 for UCS and 0.831 for E, outperforming individual models. Parameter optimization using grid search and k-fold cross-validation enhanced performance, while sensitivity analysis validated input parameter significance. The results demonstrate the stacking method's superiority in rock property prediction, offering a reliable alternative to direct laboratory testing [1].

It seems that the simultaneous use of statistical methods and artificial intelligence (AI)-based techniques can provide better results in predicting the mechanical parameters of rocks. Therefore, this study develops predictive models for sandstones' uniaxial compressive strength (UCS) and elastic modulus (E) by analyzing physical and mechanical properties through both statistical methods (i.e., correlation analysis, covariance determination, and regression modeling) and artificial intelligence-based techniques (i.e., Artificial Neural Networks, K-Nearest Neighbors, and Random Forest). The research systematically compares these traditional and modern approaches, evaluating model performance using standard statistical metrics and sensitivity analysis to establish reliable prediction tools while assessing parameter importance. By integrating statistical analysis with machine learning algorithms, this dual-method approach provides a comprehensive framework for accurate geotechnical property prediction and offers practical insights into optimal modeling strategies for sandstone mechanical characteristics. The novelty of this study lies in its integrated and systematic framework that combines comprehensive statistical analyses (covariance determination, multiple correlation metrics, and simple regression) with a comparative evaluation of multiple artificial intelligence techniques (ANN, kNN, and RF) specifically applied to sandstones from four distinct Iranian geological formations. Unlike many previous studies that focus on either statistical approaches or a single AI method, this research simultaneously evaluates traditional and AI-based models using an identical dataset, allowing for a transparent performance comparison and robustness assessment. In addition, the study emphasizes the use of easily obtainable, non-destructive physical and mechanical parameters and provides detailed sensitivity analysis to quantify the relative importance of input variables, identifying Schmidt hammer rebound as the most influential predictor for both UCS and elasticity modulus. The formation-based interpretation of results further distinguishes this work by linking predictive performance to petro-physical variability among sandstones, thereby enhancing the practical applicability of the proposed models for site-specific geotechnical design where direct laboratory testing is limited.

## 2. Methodology

### 2.1. Research steps and materials

This research is based on three continuous steps including i) sample selection and specimen preparation, ii) laboratory investigations, and iii) desk studies and data analysis (Fig 1). The first step includes operations for selecting suitable sandstones from some geological formations outcropped in different areas of Iran (Fig 2). In this step, 20 samples of sandstones were collected from four geologic formations namely Caspian (CPS), Padeha (PDH), Fajan (FJN), and Hezardareh (HZD) (Table 1). Sample collection was carried out in accordance with national regulations. Necessary permits for rock sampling were obtained from the governorate and environmental departments of the relevant provinces prior to fieldwork. Damghan University, as the authors' institution, also granted field sampling permits. Required specimens were prepared

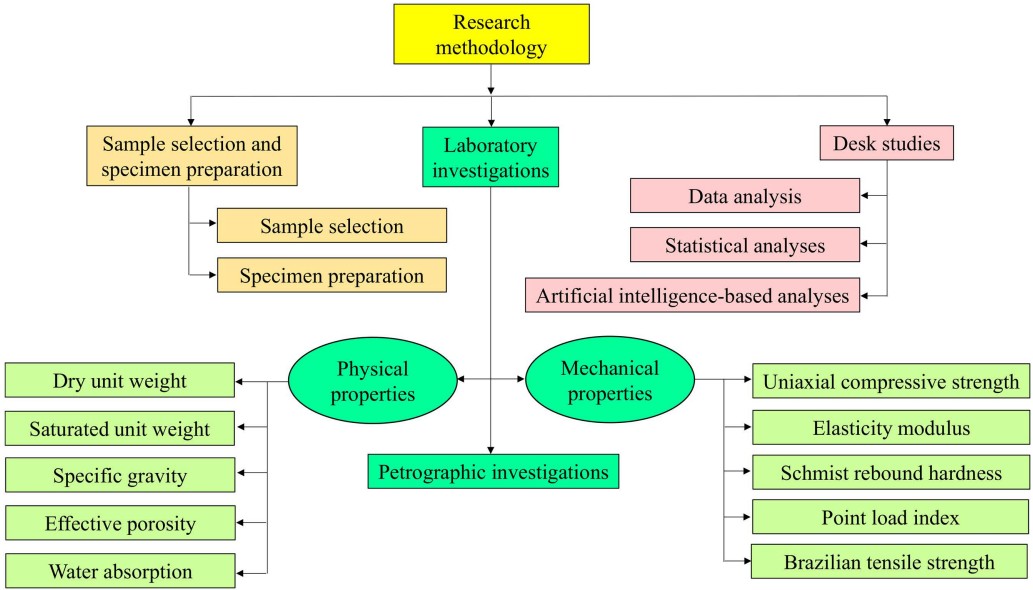

**Fig 1. Methodology flowchart of the research.**

from the selected samples rock using coring and rock cutter machines in the laboratory. The prepared specimens were washed with water to remove dust deposited on the stone surfaces during rock cutting and then have been dried at 105°C in oven. The washing and oven-drying the specimens were performed solely to remove surface dust, cutting residues, and free moisture, in accordance with ISRM (2007) recommendations, and to ensure consistent initial conditions for physical property measurements. This procedure is widely adopted in rock mechanics studies and is not expected to significantly alter the intrinsic mechanical behavior of intact sandstones, particularly for dense, low- to moderately porous materials such as those investigated here. The second step includes a comprehensive laboratory test program for evaluating physical and mechanical properties of the collected sandstones using related apparatuses. Also, one thin section for each rock sample were provided to investigate petrographic properties. The third step includes data analysis by using statistical approaches and artificial intelligence-based techniques to predict the uniaxial compressive strength (UCS) and elasticity modulus (E) from the physical and mechanical properties of the selected sandstones. The former includes covariance determination, correlation analysis, and simple regression analysis as well as the later includes Artificial Neural Network (ANN), K-Nearest Neighbors (kNN) and Random Forest (RF).

## 2.2. Test procedures

A comprehensive laboratory test program was carried out on the prepared rock specimens to assess petrographic characteristics, physical-, and mechanical properties. The petrographic investigations were done by optical microscopy studies on polished thin sections based on suggested methods of American Society for Testing and Materials (ASTM [44]) and International Society for Rock Mechanics (ISRM [45]) to ensure proper lithological identification and identify correct lithological name of the selected sandstones. The irregular shape-based method was used on the prepared specimens to determine the physical properties namely dry unit weight ($\gamma_d$), saturated unit weight ($\gamma_s$), specific gravity ($G_s$), effective porosity ($n_e$), and water absorption ($W_a$) in accordance with ASTM [44] and ISRM [45]. The Schmidt rebound hardness ($H_s$) was performed according to ISRM [45] and ASTM [46]. For this purpose, the Schmidt hammer has been used on rock blocks with the size of about $30 \times 40 \times 50$ cm to determine the $H_s$. In the point load test, cylindrical rock specimens were

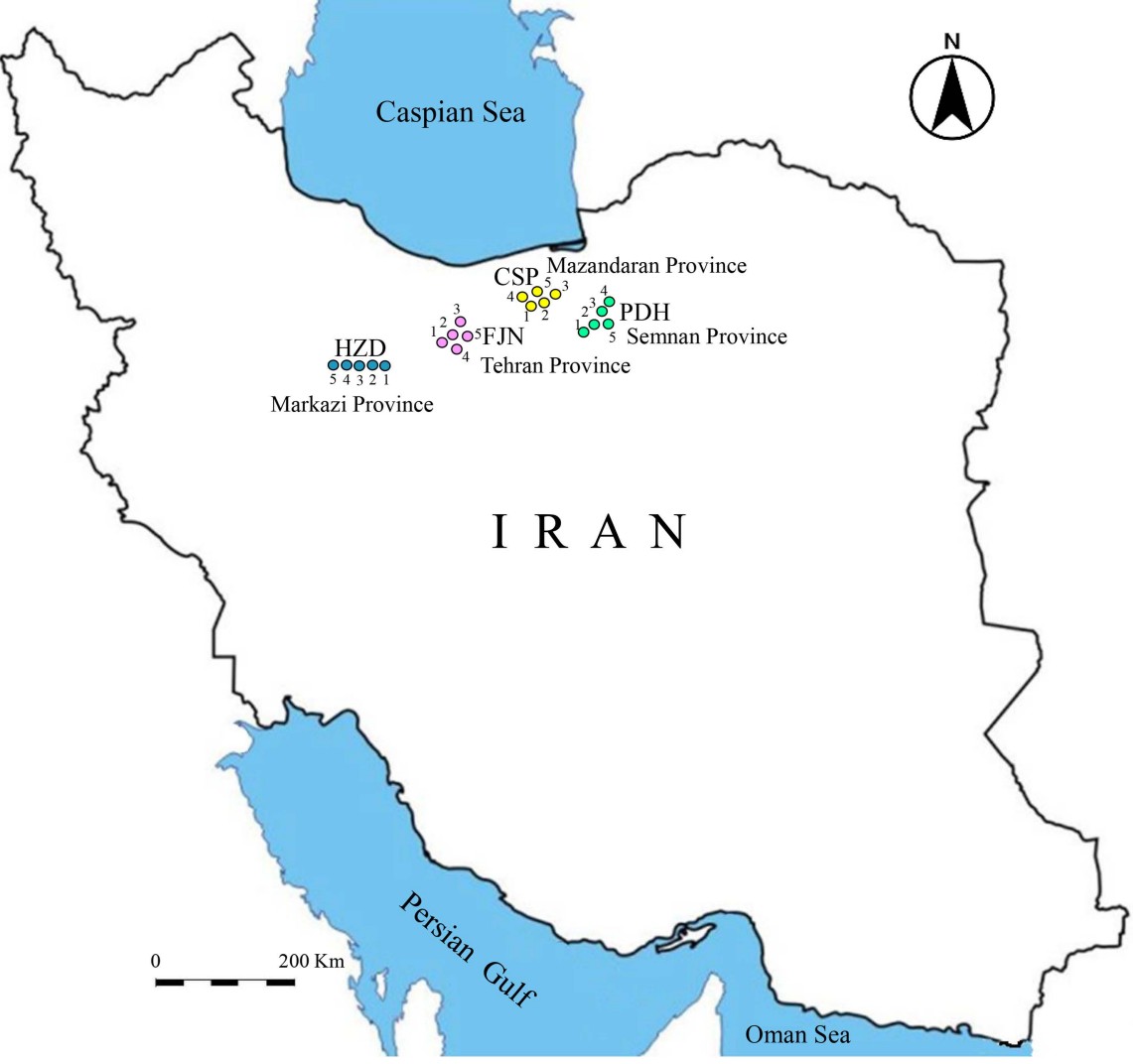

**Fig 2. Sampling locations on the general map of Iran.**

loaded between two conical platens. They failed when they developed one or more extensional planes along the line of loading. Point load index ($I_s$) is calculated using the following equation (ASTM [47]; ISRM [48]):

$$I_s = \frac{P}{D_e{}^2}$$

(1)

where P is applied force, and $D_e$ is the distance between the platens (equivalent core diameter). The point load index for a core diameter equal to 50 mm (PLI) is calculated from the following expression:

$$PLI = I_s \left(\frac{D_e}{50}\right)^2$$

(2)

**Table 1. Names of the rock samples and their information.**

| Geologic formation | Rock mark | Lithology | Province | City |
|---|---|---|---|---|
| Caspian | CSP1 | Sandstone | Mazandaran | Galoogah |
| | CSP2 | Sandstone | Mazandaran | Galoogah |
| | CSP3 | Sandstone | Mazandaran | Galoogah |
| | CSP4 | Sandstone | Mazandaran | Galoogah |
| | CSP5 | Sandstone | Mazandaran | Galoogah |
| Padeha | PDH1 | Sandstone | Semnan | Damghan |
| | PDH2 | Sandstone | Semnan | Damghan |
| | PDH3 | Sandstone | Semnan | Damghan |
| | PDH4 | Sandstone | Semnan | Damghan |
| | PDH5 | Sandstone | Semnan | Damghan |
| Fajan | FJN1 | Sandstone | Tehran | Tehran |
| | FJN2 | Sandstone | Tehran | Tehran |
| | FJN3 | Sandstone | Tehran | Tehran |
| | FJN4 | Sandstone | Tehran | Tehran |
| | FJN5 | Sandstone | Tehran | Tehran |
| Hezardareh | HZD1 | Sandstone | Markazi | Saveh |
| | HZD2 | Sandstone | Markazi | Saveh |
| | HZD3 | Sandstone | Markazi | Saveh |
| | HZD4 | Sandstone | Markazi | Saveh |
| | HZD5 | Sandstone | Markazi | Saveh |

The Brazilian tensile strength (BTS) is an undirected method to determine the tensile strength of rocks. In this test, a compressive load is vertically applied by two arch jaws, and tensile stress is perpendicular to compressive load axis. Forasmuch as the load axis is always upright, tensile stress can be calculated in the horizontal direction. Fig 3 shows the steps of performing the Brazilian tensile strength test. In this method, the BTS can be calculated from the following formula (ASTM [49]; ISRM [50]):

$$BTS = \frac{2P}{\pi Dt} = 0.636 \frac{P}{Dt} \tag{3}$$

where P is the maximum load recorded during the test. D is the diameter, and t is the thickness of the specimen.

The uniaxial compressive strength test is performed on cylindrical specimens based on ASTM [51] and ISRM [52]. In this test, the rock specimen is subjected to axial load until failure. In this case, the UCS is calculated from the following equation:

$$UCS = \frac{4P}{\pi D^2} \tag{4}$$

where P is the maximum load recorded at the moment of failure and D is the specimen diameter. The elasticity modulus (E) could be calculated by conducting the UCS test from the following equation:

$$E = \frac{\Delta \sigma}{\Delta \varepsilon} \tag{5}$$

where $\Delta\sigma$ and $\Delta\varepsilon$ are the stress and strain changes, respectively, at 50% of the specimen deformation.

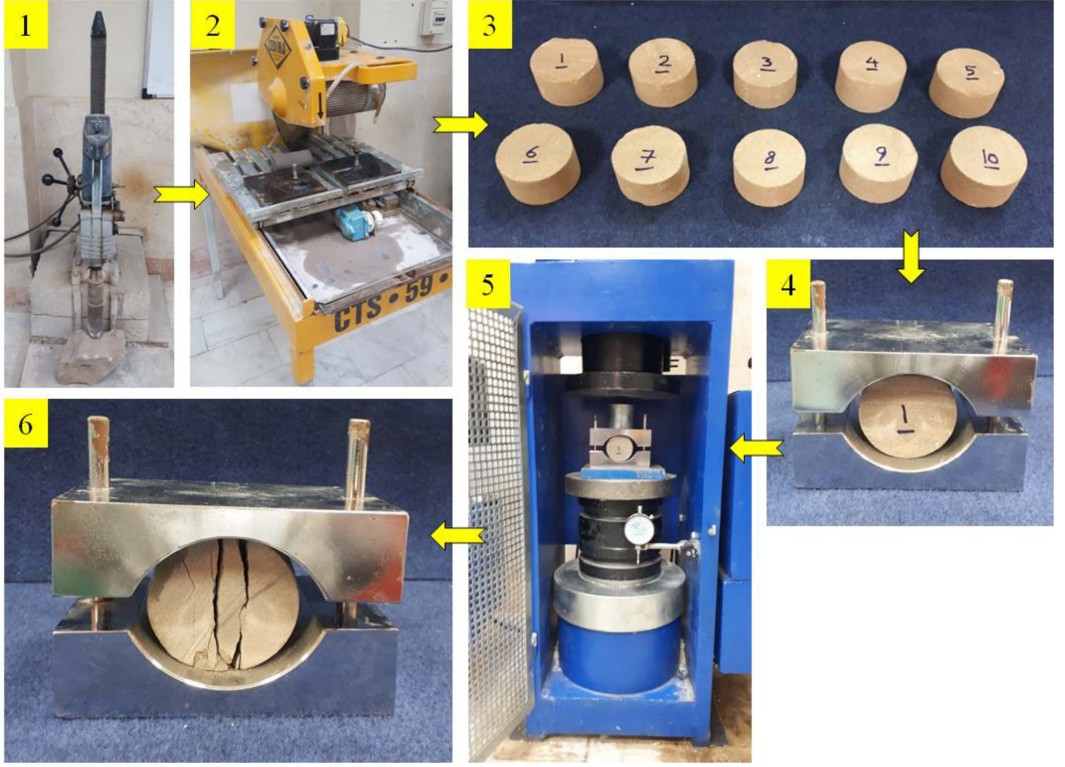

**Fig 3. Steps of performing the Brazilian tensile strength test on the sample of CSP1 as a representative sample, 1) Core extraction from the sample using a rock coring machine, 2) Cutting the head and bottom of the core using a rock cutter machine, 3) Specimens prepared for testing, 4) Placing the Specimens between the test jaws, 5) Placing the jaws inside the jack to apply pressure and perform the test, and 6) Sample failure pattern after performing the Brazilian tensile strength test.**

## 2.3. Rock properties

After performing the considered laboratory experiments on the selected sandstone samples, the obtained results were summarized in Table 2. The provided dataset highlights significant variations in the engineering properties of rock samples from four formations (CSP, PDH, FJN, HZD), offering insights into their mechanical behavior and suitability for construction applications. The PDH formation stands out with the highest dry unit weight ($\gamma_d = 25.52–25.99\,kN/m^3$), specific gravity ($G_s = 2.70–2.74$), and saturated unit weight ($\gamma_{sat} = 25.99–26.19\,kN/m^3$), correlating with its superior strength parameters, including the highest average UCS (97.21 MPa), BTS (4.11 MPa), PLI (4.37 MPa), and E (54.82 GPa), alongside low effective porosity ($n_e = 2.48–4.84\%$) and water absorption ($W_a = 0.94–1.86\%$). In contrast, the HZD formation exhibits the weakest and most porous characteristics, with the lowest $\gamma_d$ (23.34–23.85 kN/m³), $G_s$ (2.52–2.57), UCS (60.25 MPa), BTS (2.50 MPa), and E (41.87 GPa), coupled with high $n_e$ (8.02–10.16%) and $W_a$ (3.55–4.27%). The CSP formation shows intermediate properties, with moderate $\gamma_d$ (24.54–25.28 kN/m³), $G_s$ (2.64–2.67), and strength (UCS = 82.4–98.13 MPa), but higher porosity variability ($n_e = 3.02–5.61\%$) and $W_a$ (1.17–2.24%). The FJN formation, while similar in $\gamma_d$ (25.25–26.10 kN/m³) and $G_s$ (2.59–2.74) to CSP, distinguishes itself with the lowest $W_a$ (0.46–0.61%) and $n_e$ (2.07–2.71%), resulting in consistent but slightly lower strength (UCS = 85.14–96.03 MPa) and stiffness (E = 49.80–54.52 GPa) compared to PDH. Overall, the data underscores strong relationships between physical properties

**Table 2. Engineering characteristics of the studied rock samples.**

| Rock mark | $\gamma_d$ (kN/m³) | $\gamma_{sat}$ (kN/m³) | $G_s$ | $n_e$ (%) | $W_a$ (%) | Hs | PLI (MPa) | BTS (MPa) | UCS (MPa) | E (GPa) |
|---|---|---|---|---|---|---|---|---|---|---|
| CSP1 | 24.93 | 25.17 | 2.65 | 4.44 | 1.32 | 40.00 | 4.07 | 3.91 | 92.62 | 51.38 |
| CSP2 | 25.28 | 25.58 | 2.67 | 3.02 | 2.24 | 41.60 | 4.13 | 3.80 | 98.13 | 53.79 |
| CSP3 | 25.17 | 25.50 | 2.66 | 3.78 | 1.47 | 40.20 | 4.20 | 3.91 | 93.53 | 51.68 |
| CSP4 | 24.54 | 25.09 | 2.64 | 5.61 | 1.17 | 37.90 | 3.70 | 3.63 | 82.40 | 48.25 |
| CSP5 | 25.21 | 25.57 | 2.66 | 3.38 | 1.76 | 40.30 | 4.28 | 4.03 | 93.99 | 51.83 |
| PDH1 | 25.69 | 26.07 | 2.72 | 4.29 | 1.17 | 41.00 | 4.37 | 4.11 | 97.26 | 54.88 |
| PDH2 | 25.70 | 26.11 | 2.72 | 3.70 | 1.41 | 41.17 | 4.37 | 4.11 | 98.05 | 55.13 |
| PDH3 | 25.52 | 25.99 | 2.70 | 4.84 | 0.94 | 38.67 | 4.08 | 3.92 | 86.69 | 51.39 |
| PDH4 | 25.87 | 26.12 | 2.73 | 3.08 | 1.64 | 42.17 | 4.58 | 4.21 | 100.90 | 56.65 |
| PDH5 | 25.99 | 26.19 | 2.74 | 2.48 | 1.86 | 42.42 | 4.58 | 4.26 | 104.16 | 57.03 |
| FJN1 | 26.10 | 26.65 | 2.74 | 2.07 | 0.61 | 40.09 | 4.22 | 3.93 | 96.03 | 54.52 |
| FJN2 | 25.70 | 26.16 | 2.69 | 2.23 | 0.57 | 39.40 | 3.90 | 3.67 | 89.91 | 52.48 |
| FJN3 | 25.43 | 25.55 | 2.62 | 2.53 | 0.48 | 38.70 | 3.82 | 3.59 | 86.83 | 51.44 |
| FJN4 | 25.25 | 25.26 | 2.59 | 2.71 | 0.46 | 37.60 | 4.01 | 3.93 | 85.14 | 49.80 |
| FJN5 | 25.58 | 25.70 | 2.64 | 2.30 | 0.50 | 39.00 | 3.84 | 3.61 | 88.14 | 51.89 |
| HZD1 | 23.52 | 24.40 | 2.55 | 8.86 | 3.96 | 32.30 | 2.72 | 2.56 | 61.90 | 42.03 |
| HZD2 | 23.85 | 24.72 | 2.57 | 8.02 | 4.27 | 33.80 | 2.84 | 2.56 | 67.27 | 44.21 |
| HZD3 | 23.38 | 24.36 | 2.53 | 9.63 | 3.65 | 30.80 | 2.50 | 2.40 | 56.78 | 39.86 |
| HZD4 | 23.50 | 24.38 | 2.54 | 9.16 | 3.82 | 31.70 | 2.63 | 2.50 | 59.82 | 42.16 |
| HZD5 | 23.34 | 24.27 | 2.52 | 10.16 | 3.55 | 30.40 | 2.61 | 2.58 | 55.46 | 41.08 |

(e.g., density, porosity) and mechanical performance (e.g., strength, elasticity), with PDH being the most robust, FJN the most moisture-resistant, CSP moderately variable, and HZD requiring significant reinforcement for practical use. The relatively narrow range of physical and mechanical properties observed within samples from the same formation reflects their similar depositional environment, mineralogical composition, and diagenetic history, which is typical for sandstones originating from a single geological unit. From a statistical perspective, this homogeneity enhances internal consistency and reduces noise, allowing more reliable identification of intrinsic relationships between input parameters and UCS and elasticity modulus, as evidenced by the high correlation coefficients and low prediction errors. However, it is acknowledged that limited intra-formation variability may constrain the extrapolation of the derived regression equations beyond the studied property ranges. Accordingly, the developed equations and AI models are most suitable for sandstones with comparable lithological characteristics and physical property ranges, and their direct application to other rock types or highly heterogeneous sandstones should be undertaken with caution. Table 3 provided the statistical parameters for the evaluated engineering characteristics of the studied rock samples and Fig 4 shows distribution histograms of the evaluated engineering characteristics of the studied sandstones samples.

## 3. Statistical analyses

Many researchers have correlated different engineering properties of rocks to assess their relations with each other [53]. In the present research, statistical analyses were carried out for comparing the obtained results of the uniaxial compressive strength (UCS) and elasticity modulus (E) from the physical and mechanical properties of the tested sandstones and determining their correlation.

 

**Table 3. Statistical parameters for the evaluated engineering characteristics of the studied rock samples.**

| Parameter | $\gamma_d$ (kN/m³) | $\gamma_{sat}$ (kN/m³) | $G_s$ | $n_e$ (%) | $W_a$ (%) | Hs | PLI (MPa) | BTS (MPa) | UCS (MPa) | E (GPa) |
|---|---|---|---|---|---|---|---|---|---|---|
| Minimum | 23.34 | 24.27 | 2.52 | 2.07 | 0.46 | 30.40 | 2.50 | 2.40 | 55.46 | 39.86 |
| Maximum | 26.10 | 26.65 | 2.74 | 10.16 | 4.27 | 42.42 | 4.58 | 4.26 | 104.16 | 57.03 |
| Average | 24.98 | 25.44 | 2.64 | 4.81 | 1.84 | 37.96 | 3.77 | 3.56 | 84.75 | 50.07 |
| Median | 25.27 | 25.56 | 2.66 | 3.74 | 1.44 | 39.20 | 4.04 | 3.85 | 89.03 | 51.56 |
| Standard deviation | 0.94 | 0.72 | 0.07 | 2.76 | 1.29 | 3.91 | 0.70 | 0.64 | 15.63 | 5.35 |
| Variance | 0.88 | 0.51 | 0.01 | 7.61 | 1.67 | 15.31 | 0.49 | 0.41 | 244.36 | 28.66 |
| Sig. (Kolmogorov-Smirnov) | 0.006 | 0.20 | 0.20 | 0.29 | 0.046 | 0.011 | 0.008 | 0.001 | 0.021 | 0.020 |
| Sig. (Shapiro-Wilk) | 0.005 | 0.151 | 0.139 | 0.002 | 0.007 | 0.005 | 0.003 | 0.001 | 0.007 | 0.022 |

### 3.1. Covariance determination

Covariance is a measure of the relationship between two random variables, in statistics. The covariance indicates the relation between the two variables and helps to know if the two variables vary together. The covariance between two random variables of X and Y can be denoted as Cov(X, Y) [54]:

$$Cov(X, Y) = \frac{1}{n} \sum_{i=1}^{n} (X_i - \overline{X})(Y_i - \overline{Y})$$

(6)

where $X_i$ and $Y_i$ are the values of the X and Y variables, $\overline{X}$ and $\overline{Y}$ are the mean of the X and Y variables, and n is the number of data points. In the present research, X is considered UCS and E, and Y is considered dry unit weight ($\gamma_d$), specific gravity ($G_s$), effective porosity ($n_e$), Schmidt rebound hardness ($H_s$), point load index (PLI), and Brazilian tensile strength (BTS), so the covariances between the parameters for the studied rocks were calculated in Microsoft Excel 2024 (Table 4). The obtained results indicates that except $n_e$, the other parameters are directly correlated to the UCS and E because their covariances have positive values.

### 3.2. Correlation analysis

Correlation analysis is a statistical approach to determine relationship between the dependent and independent variables. In the present research, the correlation analysis is carried out by Microsoft Excel 2024, IBM SPSS 26, and Grapher 22.1.333 software and the correlations between the rock parameters are shown in Figs 5,6. There are direct linear relationships between the UCS and E with dry unit weight ($\gamma_d$), specific gravity ($G_s$), Schmidt rebound hardness ($H_s$), point load index (PLI), and Brazilian tensile strength (BTS), while inverse linear correlations between the UCS and E with effective porosity ($n_e$). After extraction the linear relation between UCS and E with other physical and mechanical properties, correlations between the experimental and predicted values of UCS and E were developed that they are presented in Figs 7,8. According to the results, the predicted values of UCS and E are generally equal to the experimental values and the obtained line is approximately fitted the 45° line (y = x). This means that there are good correlations between the predicted and experimental values of UCS and E. Figs 9,10 comprise the values of experimental UCS and E with their predicted values from the rock physical and mechanical properties. They well show that these values are coordinate and the values of the experimental and predicted values of UCS and E are related to each other. Also, to better understand the relationship between the values of UCS and E with the other physical and mechanical properties, several 3D views of the correlations among experimental UCS and E with $\gamma_d$, $n_e$, PLI, and BTS are presented in Fig 11. All parts of this figure are very similar to each other and this shows the close relationship between the values of the parameters.

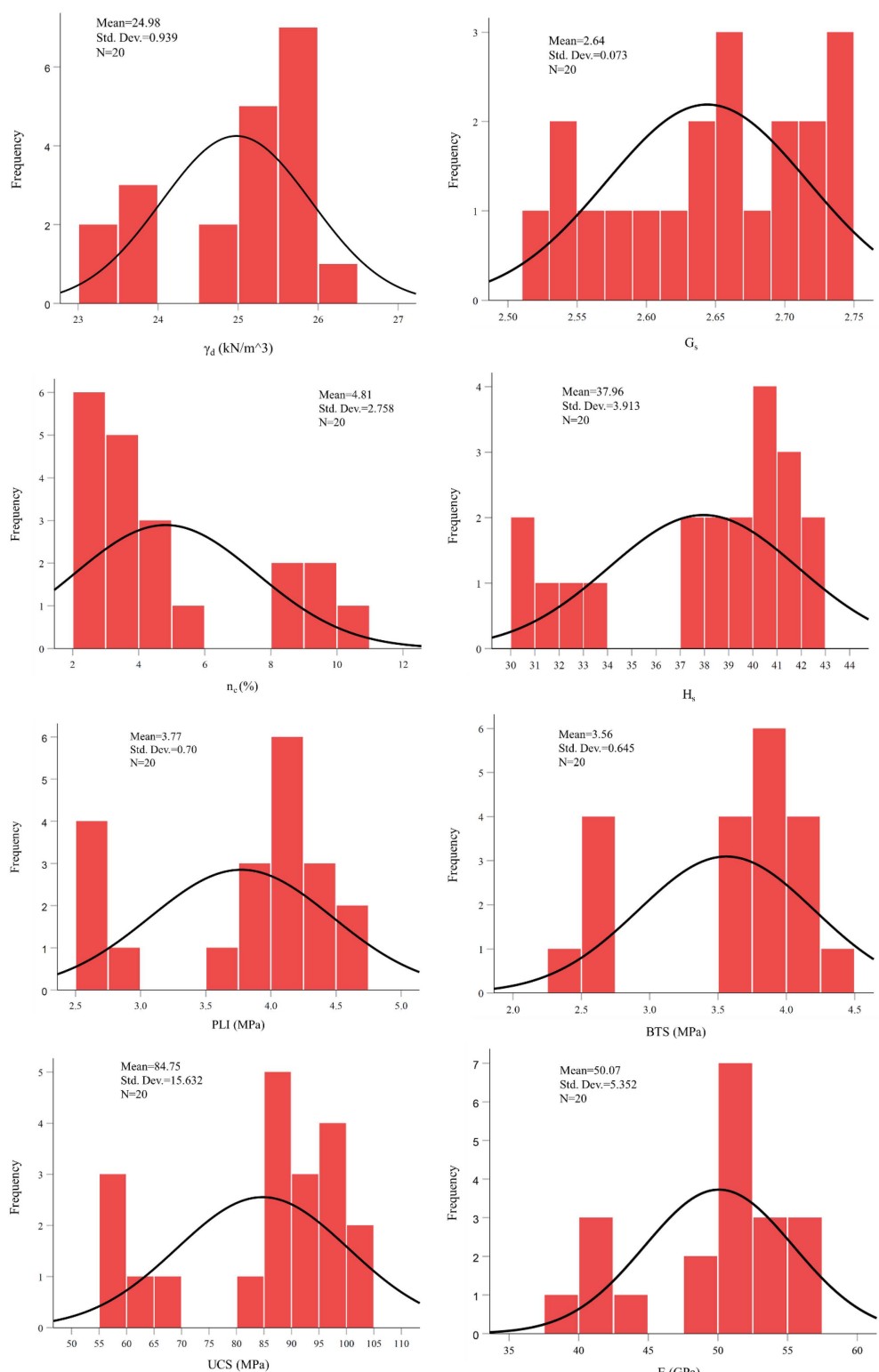

**Fig 4. Distribution histograms of the evaluated engineering characteristics of the studied rock samples.**

**Table 4. The values of covariance between the rock parameters.**

| | $\gamma_d$ | $G_s$ | $n_e$ (%) | $H_s$ | PLI (MPa) | BTS (MPa) |
|---|---|---|---|---|---|---|
| UCS | 13.21 | 1.01 | −37.61 | 57.95 | 10.25 | 9.25 |
| E | 4.64 | 0.35 | −12.86 | 19.55 | 3.47 | 3.13 |

To ensure the correlations obtained between the above-mentioned parameters, the Pearson, Spearman, and Kendall correlation coefficients were calculated in IBM SPSS 26. These are correlation coefficients used to statistical measure of the relationship strength between paired data. In Pearson method, the values of the statistical parameters, including the coefficients of correlation and determination ($R$ and $R^2$) can be calculated from the equations below [54]:

$$R = \frac{n\Sigma x_i y_i - (\Sigma x_i)(\Sigma y_i)}{\sqrt{n(\Sigma x_i^2) - (\Sigma x_i)^2}\sqrt{n(\Sigma y_i^2) - (\Sigma y_i)^2}}$$

(7)

$$R^2 = 1 - \frac{\sum_i (y_i - y_i')^2}{\sum_i (y_i - \bar{y})^2}$$

(8)

where x and y are independent and dependent variables, respectively. n is the total number of data (20 rock samples). $y_i$ and $\bar{y}$ are the predicted and mean values of the y, respectively. The correlation matrix (Table 5) reveals strong, statistically significant relationships ($p < 0.01$) among geotechnical parameters, with dry unit weight ($\gamma_d$), saturated unit weight ($\gamma_{sat}$), and specific gravity ($G_s$) showing high positive correlations ($R > 0.9$) with mechanical strength indicators (PLI, BTS, UCS, E) and Schmidt hammer rebound ($H_s$), indicating that denser, less porous materials exhibit greater strength and stiffness. Conversely, porosity ($n_e$) and water absorption ($W_a$) display strong negative correlations ($R < −0.75$) with these mechanical properties, suggesting that increased void space and moisture content reduce rock strength. The strongest interdependencies exist between $H_s$, PLI, UCS, and Young's modulus (E), with near-perfect correlations ($R > 0.98$), emphasizing their reliability as complementary measures of rock strength.

In Spearman method, the calculation of the rank correlation coefficient and subsequent significance testing of it are as follow [54]:

$$r_s = 1 - \frac{6\sum d_i^2}{n(n^2 - 1)}$$

(9)

where $r_s$ is Spearman rank correlation coefficient, $d_i$ is the difference between the ranks of corresponding variables, and n is number of observations (20 rock samples). The value of the Spearman's correlation coefficient for the studied rocks is presented in Table 6. The correlation analysis reveals that dry unit weight ($\gamma_d$) and saturated unit weight ($\gamma_{sat}$) exhibit very strong positive relationships ($R > 0.9$, $p < 0.01$) with specific gravity ($G_s$) and mechanical properties (PLI, BTS, UCS, E), indicating that denser materials possess greater strength and stiffness. Porosity ($n_e$) shows significant negative correlations ($R \approx −0.5$ to $−0.8$, $p < 0.05$) with these properties, confirming that higher void content weakens the material. Water absorption ($W_a$) displays weaker, often insignificant relationships, except with ne ($R = 0.664$, $p < 0.01$). Notably, Schmidt hammer rebound ($H_s$) and UCS demonstrate near-perfect correlation ($R = 0.994$, $p < 0.01$), validating $H_s$ as a reliable indirect strength indicator. Young's modulus (E) strongly correlates with $\gamma_d$, $\gamma_{sat}$, $G_s$, and UCS ($R > 0.9$, $p < 0.01$), reinforcing the interdependence of density, strength, and stiffness in geotechnical behavior.

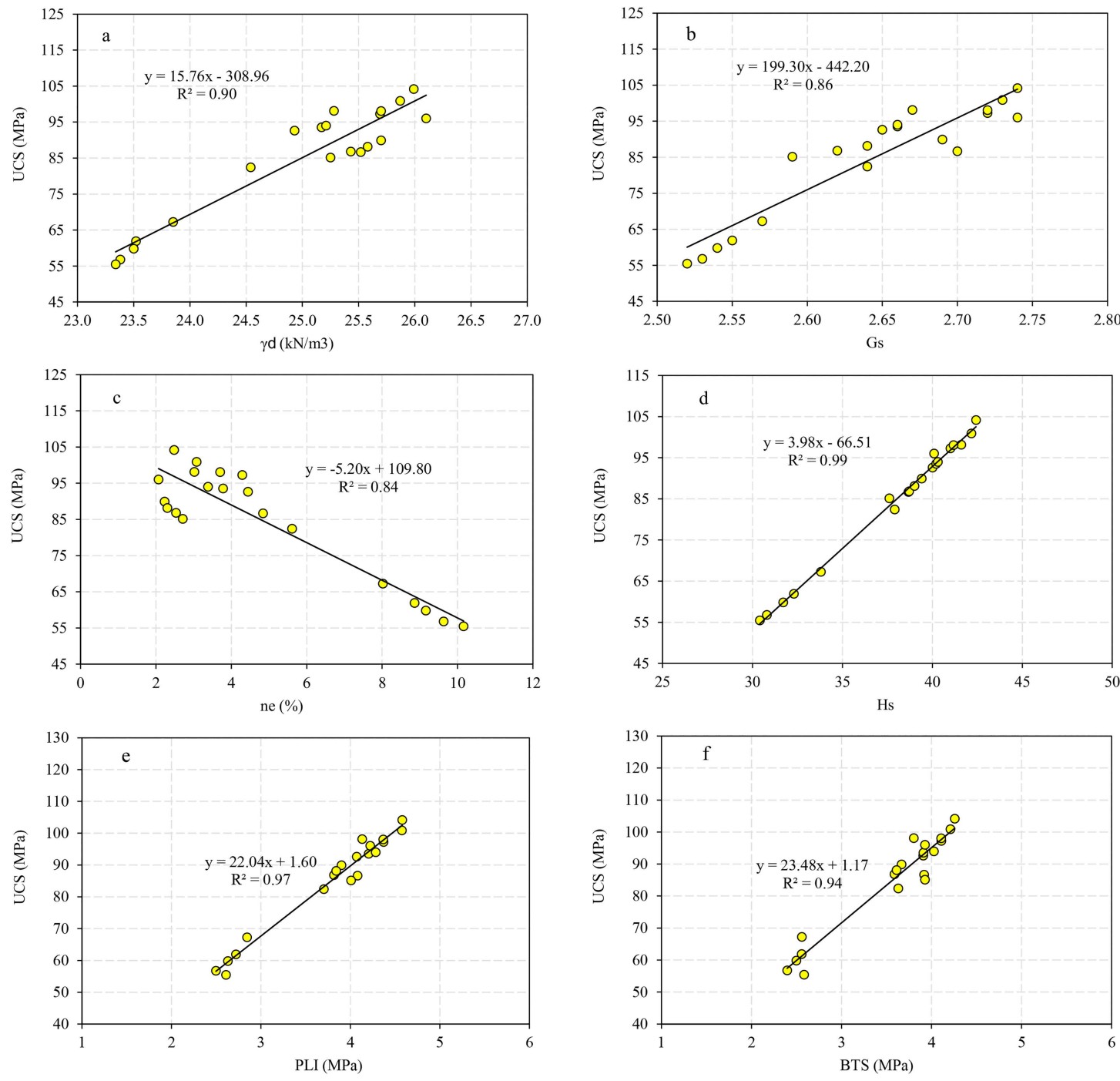

**Fig 5. Correlations between the values of experimental UCS and a) γd, b) Gs, c) ne, d) Hs, e) PLI, and f) BTS.**

The Kendall's method is applied to the ranks of the data to determine the strength of the relationship between two variables. If $(x_1, y_1)$, $(x_2, y_2)$, …, $(x_n, y_n)$ be a set of joint observations from two random variables X and Y respectively, such that all the values of $(x_i)$ and $(y_i)$ are unique. Any pair of observations $(x_i, y_i)$ and $(x_j, y_j)$ are concordant if the ranks for both

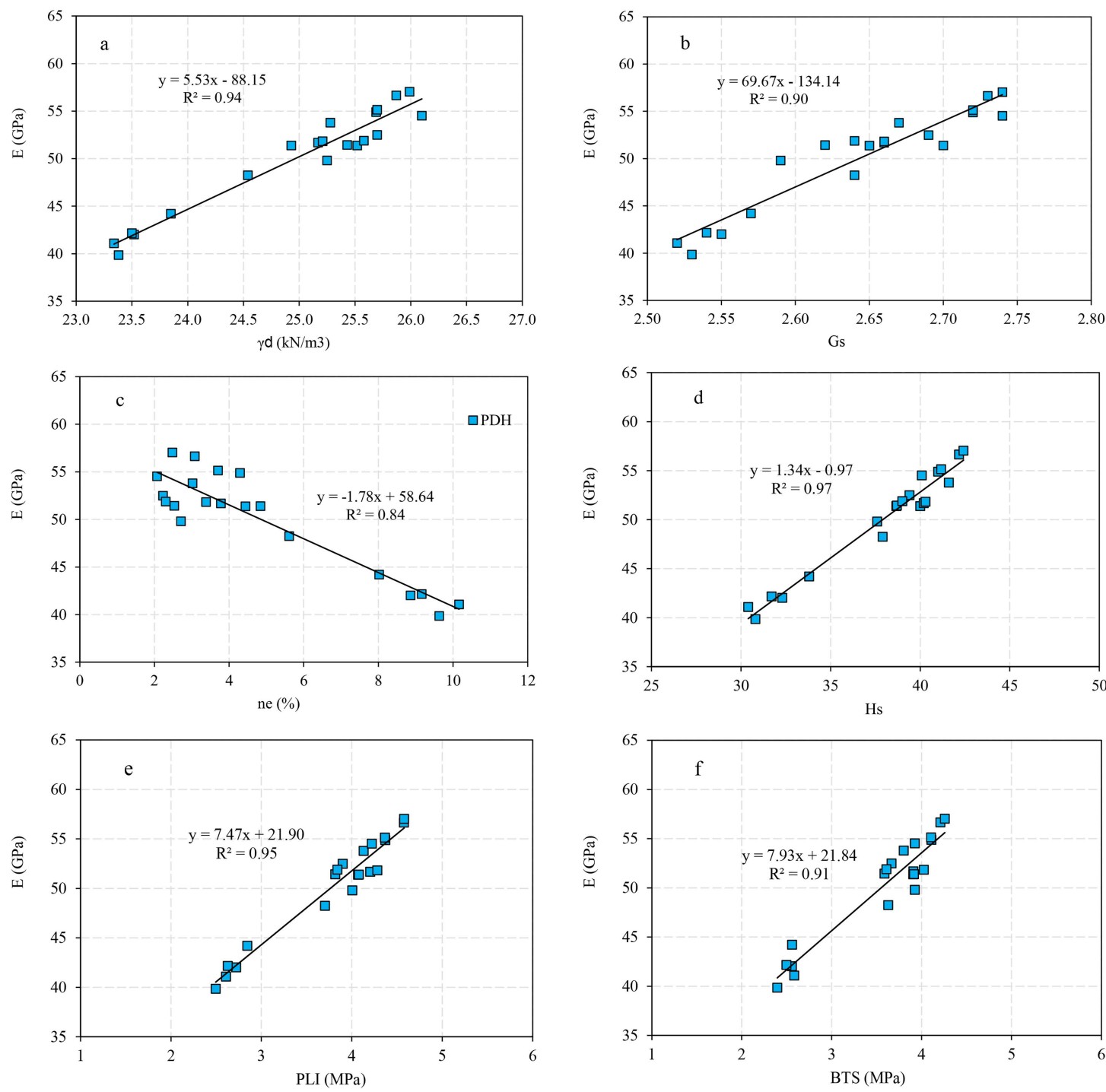

**Fig 6. Correlations between the values of experimental E and a) γd, b) Gs, c) ne, d) Hs, e) PLI, and f) BTS.**

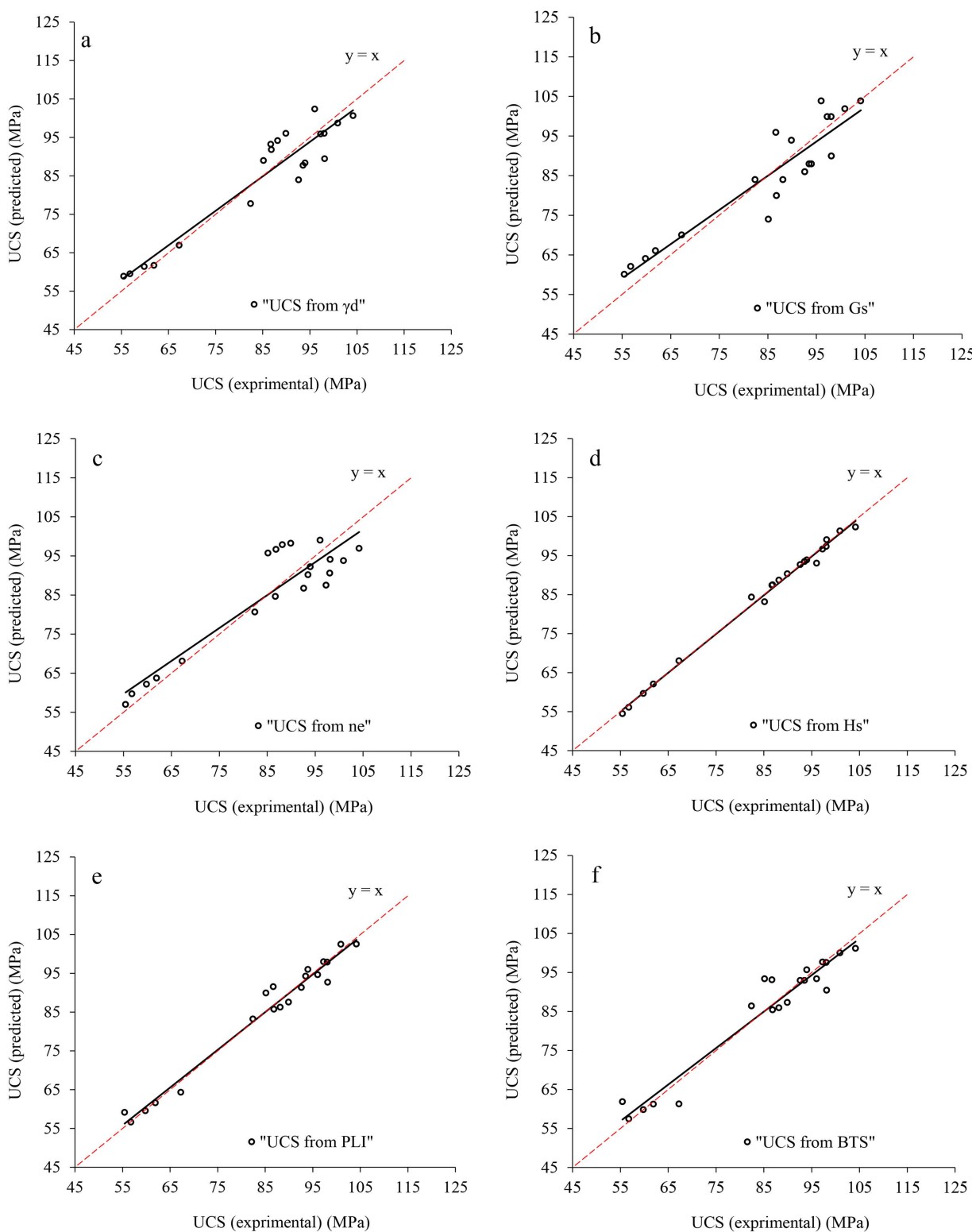

**Fig 7. Correlations between the experimental and predicted values of UCS.**

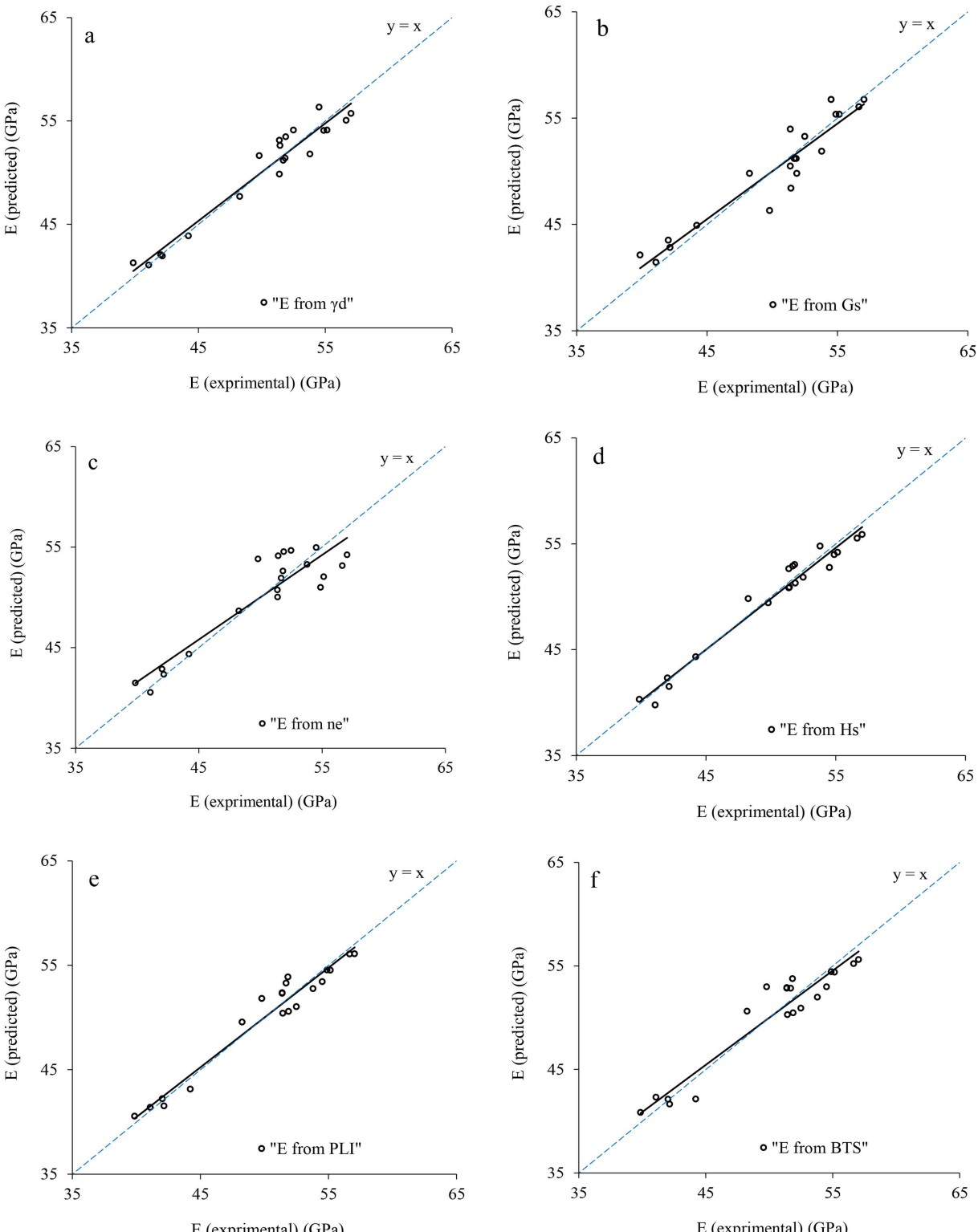

**Fig 8. Correlations between the experimental and predicted values of E.**

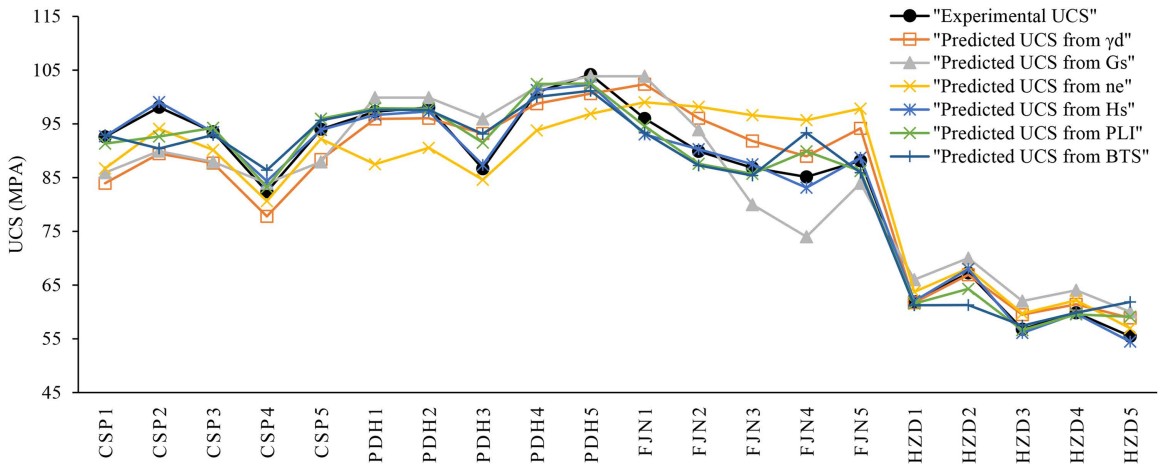

**Fig 9. Comparison of the experimental and predicted values of UCS.**

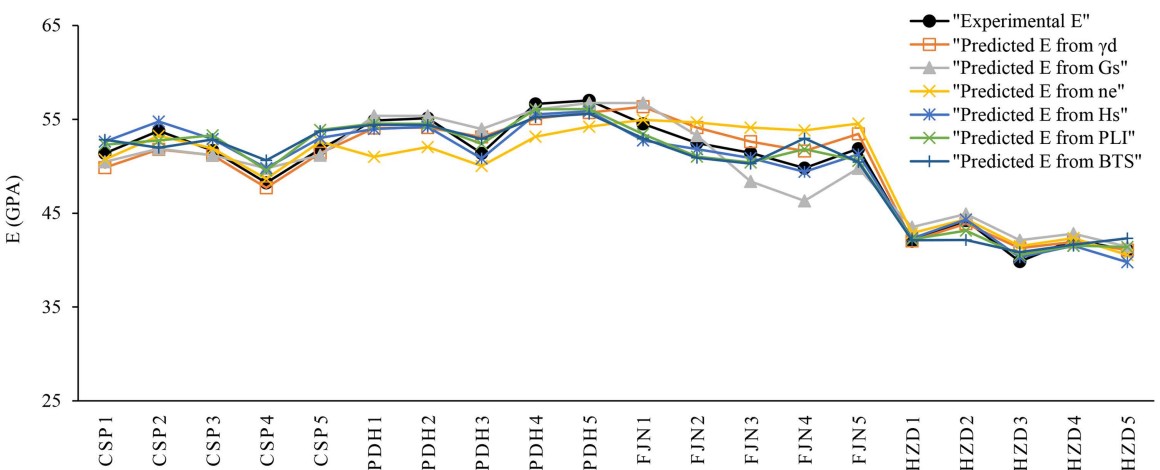

**Fig 10. Comparison of the experimental and predicted values of E.**

elements agree: that is, if both xi > xj and $y_i > y_j$ or if both $x_i < x_j$ and $y_i < y_j$. They are discordant, if $x_i > x_j$ and $y_i < y_j$ or if $x_i < x_j$ and $y_i > y_j$. If $x_i = x_j$ or $y_i = y_j$, the pair is neither concordant, nor discordant. For n observations, (i.e., (i, j) ∈ {1, . . ., n}2) the number of concordant C, discordant D, tied pairs T in X, and tied pairs U in Y is denoted as follows [55]:

$$C = |\{(i, j)|x_i < x_j \text{ and } y_i < y_j\}| \tag{10}$$

$$D = |\{(i, j)|x_i < x_j \text{ and } y_i > y_j\}| \tag{11}$$

$$T = |\{(i, j)|x_i = x_j\}| \tag{12}$$

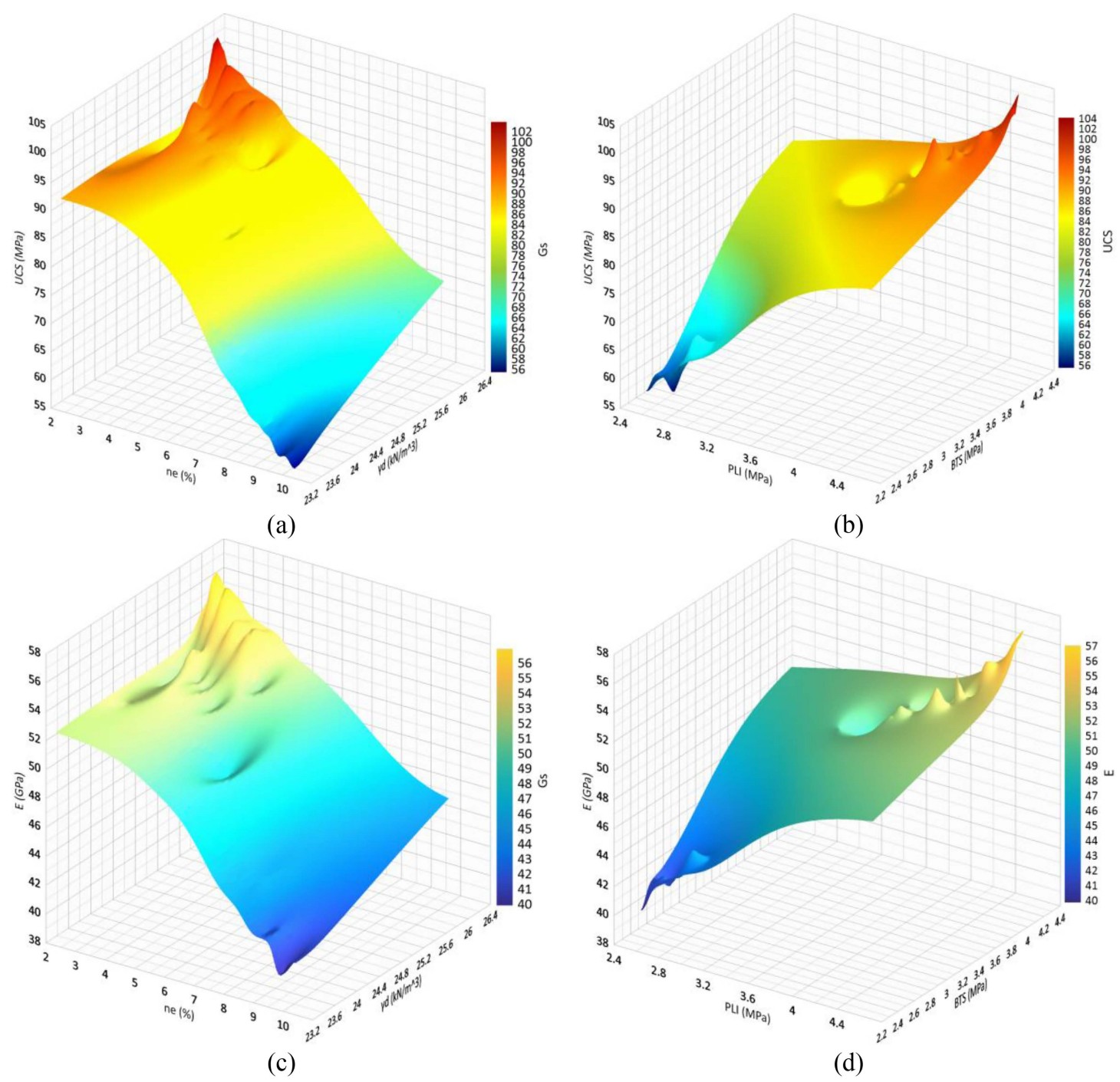

**Fig 11. 3D view of the correlations among a) experimental UCS, γd and ne, b) UCS, PLI and BTS, c) E, γd and ne, and b) E, PLI and BTS.**

   

**Table 5. Parameters of Pearson correlation analysis.**

| | | $\gamma_d$ (kN/m³) | $\gamma_{sat}$ (kN/m³) | $G_s$ | $n_e$ (%) | $W_a$ (%) | $H_s$ | PLI (MPa) | BTS (MPa) | UCS (MPa) | E (GPa) |
|---|---|---|---|---|---|---|---|---|---|---|---|
| $\gamma_d$ (kN/m³) | R | 1 | 0.971** | 0.924** | −0.953** | −0.852** | 0.937** | 0.946** | 0.932** | 0.947** | 0.971** |
| | Sig. (2-tailed) | | 0.000 | 0.000 | 0.000 | 0.000 | 0.000 | 0.000 | 0.000 | 0.000 | 0.000 |
| $\gamma_{sat}$ (kN/m³) | R | 0.971** | 1 | 0.959** | −0.877** | −0.772** | 0.886** | 0.893** | 0.870** | 0.899** | 0.937** |
| | Sig. (2-tailed) | 0.000 | | 0.000 | 0.000 | 0.000 | 0.000 | 0.000 | 0.000 | 0.000 | 0.000 |
| $G_s$ | R | 0.924** | 0.959** | 1 | −0.805** | −0.679** | 0.923** | 0.920** | 0.896** | 0.929** | 0.948** |
| | Sig. (2-tailed) | 0.000 | 0.000 | | 0.000 | 0.001 | 0.000 | 0.000 | 0.000 | 0.000 | 0.000 |
| $n_e$ (%) | R | −0.953** | −0.877** | −0.805** | 1 | 0.878** | −0.913** | −0.903** | −0.892** | −0.918** | −0.916** |
| | Sig. (2-tailed) | 0.000 | 0.000 | 0.000 | | 0.000 | 0.000 | 0.000 | 0.000 | 0.000 | 0.000 |
| $W_a$ (%) | R | −0.852** | −0.772** | −0.679** | 0.878** | 1 | −0.751** | −0.785** | −0.821** | −0.754** | −0.759** |
| | Sig. (2-tailed) | 0.000 | 0.000 | 0.001 | 0.000 | | 0.000 | 0.000 | 0.000 | 0.000 | 0.000 |
| $H_s$ | R | .0937** | 0.886** | 0.923** | −0.913** | −0.751** | 1 | 0.981** | 0.962** | 0.997** | 0.983** |
| | Sig. (2-tailed) | 0.000 | 0.000 | 0.000 | 0.000 | 0.000 | | 0.000 | 0.000 | 0.000 | 0.000 |
| PLI (MPa) | R | 0.946** | 0.893** | 0.920** | −0.903** | −0.785** | 0.981** | 1 | 0.993** | 0.986** | 0.976** |
| | Sig. (2-tailed) | 0.000 | 0.000 | 0.000 | 0.000 | 0.000 | 0.000 | | 0.000 | 0.000 | 0.000 |
| BTS (MPa) | R | 0.932** | 0.870** | 0.896** | −0.892** | −.821** | .962** | 0.993** | 1 | 0.967** | 0.954** |
| | Sig. (2-tailed) | 0.000 | 0.000 | 0.000 | 0.000 | 0.000 | 0.000 | 0.000 | | 0.000 | 0.000 |
| UCS (MPa) | R | 0.947** | 0.899** | 0.929** | −0.918** | −0.754** | 0.997** | 0.986** | 0.967** | 1 | 0.988** |
| | Sig. (2-tailed) | 0.000 | 0.000 | 0.000 | 0.000 | 0.000 | 0.000 | 0.000 | 0.000 | | 0.000 |
| E (GPa) | R | 0.971** | 0.937** | 0.948** | −0.916** | −0.759** | 0.983** | 0.976** | 0.954** | 0.988** | 1 |
| | Sig. (2-tailed) | 0.000 | 0.000 | 0.000 | 0.000 | 0.000 | 0.000 | 0.000 | 0.000 | 0.000 | |

**. Correlation is significant at the 0.01 level (2-tailed)

*. Correlation is significant at the 0.05 level (2-tailed)

$$U = |\{(i, j)|y_i = y_j\}| \tag{13}$$

The following formula is used for calculating the Kendall's rank correlation coefficient [56]:

$$\tau = \frac{C - D}{\sqrt{\left[\frac{n(n-1)}{2-T}\right]\left[\frac{n(n-1)}{2-U}\right]}} \tag{14}$$

The value of the Kendall's rank correlation coefficient for the studied rocks is presented in Table 7. The correlation analysis reveals strong positive relationships between density parameters ($\gamma_d$, $\gamma_{sat}$) and mechanical properties (PLI, BTS, UCS, E), with coefficients ranging from 0.60 to 0.81 (p<0.01), indicating that higher density materials generally exhibit greater strength and stiffness. Specific gravity ($G_s$) shows particularly strong correlations with strength parameters (0.73–0.79,

**Table 6. Parameters of Spearman correlation analysis.**

| | | $\gamma_d$ (kN/m³) | $\gamma_{sat}$ (kN/m³) | $G_s$ | $n_e$ (%) | $W_a$ (%) | $H_s$ | PLI (MPa) | BTS (MPa) | UCS (MPa) | E (GPa) |
|---|---|---|---|---|---|---|---|---|---|---|---|
| $\gamma_d$ (kN/m³) | R | 1.000 | 0.985** | 0.908** | −0.836** | −0.553* | 0.778** | 0.804** | 0.768** | 0.818** | 0.925** |
| | Sig. (2-tailed) | . | 0.000 | 0.000 | 0.000 | 0.011 | 0.000 | 0.000 | 0.000 | 0.000 | 0.000 |
| $\gamma_{sat}$ (kN/m³) | R | 0.985** | 1.000 | 0.942** | −0.814** | −0.496* | 0.806** | 0.825** | 0.774** | 0.839** | 0.932** |
| | Sig. (2-tailed) | 0.000 | . | 0.000 | 0.000 | 0.026 | 0.000 | 0.000 | 0.000 | 0.000 | 0.000 |
| $G_s$ | R | 0.908** | 0.942** | 1.000 | −0.644** | −0.352 | 0.881** | 0.917** | 0.860** | 0.901** | 0.924** |
| | Sig. (2-tailed) | 0.000 | 0.000 | . | 0.002 | 0.128 | 0.000 | 0.000 | 0.000 | 0.000 | 0.000 |
| $n_e$ (%) | R | −0.836** | −0.814** | −0.644** | 1.000 | 0.664** | −0.627** | −0.576** | −0.535* | −0.668** | −0.749** |
| | Sig. (2-tailed) | 0.000 | 0.000 | 0.002 | . | 0.001 | 0.003 | 0.008 | 0.015 | .001 | 0.000 |
| $W_a$ (%) | R | −0.553* | −0.496* | −0.352 | 0.664** | 1.000 | −0.205 | −0.272 | −0.360 | −0.236 | −0.356 |
| | Sig. (2-tailed) | 0.011 | 0.026 | 0.128 | 0.001 | . | 0.385 | 0.246 | 0.118 | 0.316 | 0.124 |
| $H_s$ | R | 0.778** | 0.806** | 0.881** | −0.627** | −0.205 | 1.000 | 0.939** | 0.837** | .994** | 0.940** |
| | Sig. (2-tailed) | 0.000 | 0.000 | 0.000 | 0.003 | 0.385 | . | 0.000 | 0.000 | 0.000 | 0.000 |
| PLI (MPa) | R | .804** | 0.825** | 0.917** | −0.576** | −0.272 | 0.939** | 1.000 | 0.958** | 0.945** | 0.910** |
| | Sig. (2-tailed) | 0.000 | 0.000 | 0.000 | 0.008 | 0.246 | 0.000 | . | 0.000 | 0.000 | 0.000 |
| BTS (MPa) | R | 0.768** | 0.774** | 0.860** | −0.535* | −0.360 | 0.837** | 0.958** | 1.000 | 0.849** | 0.831** |
| | Sig. (2-tailed) | 0.000 | 0.000 | 0.000 | 0.015 | 0.118 | 0.000 | 0.000 | . | 0.000 | 0.000 |
| UCS (MPa) | R | 0.818** | 0.839** | 0.901** | −0.668** | −0.236 | 0.994** | 0.945** | 0.849** | 1.000 | 0.955** |
| | Sig. (2-tailed) | 0.000 | 0.000 | 0.000 | 0.001 | 0.316 | 0.000 | 0.000 | 0.000 | . | 0.000 |
| E (GPa) | R | 0.925** | 0.932** | 0.924** | −0.749** | −0.356 | 0.940** | 0.910** | 0.831** | 0.955** | 1.000 |
| | Sig. (2-tailed) | 0.000 | 0.000 | 0.000 | 0.000 | 0.124 | 0.000 | 0.000 | 0.000 | 0.000 | . |

**. Correlation is significant at the 0.01 level (2-tailed).

*. Correlation is significant at the 0.05 level (2-tailed).

$p < 0.01$), while porosity ($n_e$) demonstrates significant negative correlations (−0.39 to −0.61, $p < 0.05$), confirming that increased void content reduces material performance. Notably, the Schmidt hammer test ($H_s$) shows an exceptionally strong relationship with UCS ($R = 0.968$, $p < 0.01$), validating its effectiveness as a non-destructive strength indicator. Water absorption ($W_a$) exhibits weak and mostly non-significant correlations, except with porosity ($R = 0.35$, $p < 0.05$). The consistent, strong correlations between Young's modulus (E) and strength parameters (0.68–0.86, $p < 0.01$) underscore the fundamental relationship between material stiffness and strength in geotechnical behavior.

The interdependence of predictor variables was rigorously assessed using three complementary correlation matrices: Pearson, Spearman, and Kendall (Tables 5–7, respectively). The Pearson matrix revealed strong linear correlations, with Schmidt hammer rebound ($H_s$) and point load index (PLI) showing near-perfect positive relationships with UCS ($R > 0.98$). The non-parametric Spearman and Kendall matrices confirmed these robust monotonic associations while being less sensitive to potential outliers. Collectively, these matrices highlight significant multicollinearity among key predictors such as $H_s$, PLI, dry unit weight ($\gamma_d$), and Brazilian tensile strength (BTS). This strong intercorrelation validates their collective

**Table 7. Parameters of Kendall correlation analysis.**

| | | $\gamma_d$ (kN/m³) | $\gamma_{sat}$ (kN/m³) | $G_s$ | $n_e$ (%) | $W_a$ (%) | $H_s$ | PLI (MPa) | BTS (MPa) | UCS (MPa) | E (GPa) |
|---|---|---|---|---|---|---|---|---|---|---|---|
| $\gamma_d$ (kN/m³) | R | 1.000 | 0.934** | 0.816** | −0.734** | −0.296 | 0.670** | 0.653** | 0.603** | 0.702** | 0.797** |
| | Sig. (2-tailed) | . | 0.000 | 0.000 | 0.000 | 0.069 | 0.000 | 0.000 | 0.000 | 0.000 | 0.000 |
| $\gamma_{sat}$ (kN/m³) | R | 0.934** | 1.000 | 0.851** | −0.695** | −0.259 | 0.684** | 0.688** | 0.617** | 0.716** | 0.811** |
| | Sig. (2-tailed) | 0.000 | . | 0.000 | 0.000 | .112 | 0.000 | 0.000 | 0.000 | 0.000 | 0.000 |
| $G_s$ | R | 0.816** | 0.851** | 1.000 | −0.543** | −0.123 | 0.777** | 0.786** | 0.726** | 0.787** | 0.798** |
| | Sig. (2-tailed) | 0.000 | 0.000 | . | 0.001 | 0.454 | 0.000 | 0.000 | 0.000 | 0.000 | 0.000 |
| $n_e$ (%) | R | −0.734** | −0.695** | −0.543** | 1.000 | 0.354* | −0.547** | −0.444** | −0.394* | −0.579** | −0.611** |
| | Sig. (2-tailed) | 0.000 | 0.000 | 0.001 | . | 0.030 | 0.001 | 0.006 | 0.016 | 0.000 | 0.000 |
| $W_a$ (%) | R | −0.296 | −0.259 | −.0123 | 0.354* | 1.000 | 0.016 | −0.090 | −0.187 | −0.016 | −.153 |
| | Sig. (2-tailed) | 0.069 | 0.112 | 0.454 | 0.030 | . | 0.922 | 0.581 | 0.255 | 0.922 | 0.347 |
| $H_s$ | R | 0.670** | 0.684** | 0.777** | −0.547** | 0.016 | 1.000 | 0.836** | 0.702** | 0.968** | 0.832** |
| | Sig. (2-tailed) | 0.000 | 0.000 | 0.000 | 0.001 | 0.922 | . | 0.000 | 0.000 | 0.000 | 0.000 |
| PLI (MPa) | R | 0.653** | 0.688** | 0.786** | −0.444** | −0.090 | 0.836** | 1.000 | 0.861** | 0.847** | 0.815** |
| | Sig. (2-tailed) | 0.000 | 0.000 | 0.000 | 0.006 | 0.581 | 0.000 | . | 0.000 | 0.000 | 0.000 |
| BTS (MPa) | R | 0.603** | 0.617** | 0.726** | −0.394* | −0.187 | 0.702** | 0.861** | 1.000 | 0.713** | 0.681** |
| | Sig. (2-tailed) | 0.000 | 0.000 | 0.000 | 0.016 | 0.255 | 0.000 | 0.000 | . | 0.000 | 0.000 |
| UCS (MPa) | R | 0.702** | 0.716** | 0.787** | −0.579** | −0.016 | 0.968** | 0.847** | 0.713** | 1.000 | 0.863** |
| | Sig. (2-tailed) | 0.000 | 0.000 | 0.000 | 0.000 | 0.922 | 0.000 | 0.000 | 0.000 | . | 0.000 |
| E (GPa) | R | 0.797** | 0.811** | 0.798** | −0.611** | −0.153 | 0.832** | 0.815** | 0.681** | 0.863** | 1.000 |
| | Sig. (2-tailed) | 0.000 | 0.000 | 0.000 | 0.000 | 0.347 | 0.000 | 0.000 | 0.000 | 0.000 | . |

**. Correlation is significant at the 0.01 level (2-tailed)

*. Correlation is significant at the 0.05 level (2-tailed)

relevance for predicting UCS but also underscores the importance of feature selection techniques, as employed in the sensitivity analysis, to identify the most non-redundant and influential parameters for the AI models.

### 3.3. Simple regression analysis

The simple regression model is an approach to predict the dependent variable from independent variable. In linear state, the model is defined as follow [54]:

$$y = \beta_0 + \beta_1 x + \varepsilon \tag{15}$$

where y is depended variable, $\beta_0$ is the intercept, $\beta_1$ is the slope are unknown constants and $\varepsilon$ is a random error component. This analysis was carried out using regression function in Microsoft Excel 2024. The calculated statistical parameters resulted from the method on the values of physical and mechanical properties of the tested rocks are presented

in Table 8. The analysis reveals exceptionally strong predictive relationships between unconfined compressive strength (UCS) and various geotechnical parameters, with particularly notable correlations for Schmidt hammer rebound ($H_S$, R = 1.00, R² = 0.99) and point load index (PLI, R = 0.99, R² = 0.97), demonstrating near-perfect linear associations. Dry unit weight ($\gamma_d$) and specific gravity ($G_s$) also show strong correlations with both UCS (R = 0.95 and 0.93 respectively) and Young's modulus (E) (R = 0.97 and 0.95), while porosity ($n_e$) maintains slightly weaker but still highly significant relationships (R = 0.92 for both UCS and E). All correlations are statistically significant ($p < 10^{-8}$), with particularly remarkable significance levels for $H_S$-UCS ($p = 6.54 \times 10^{-22}$) and $H_S$-E ($p = 1.25 \times 10^{-14}$) relationships. The standard errors are smallest for $H_S$ predictions (1.17 for UCS, 1.02 for E), confirming $H_S$ as the most precise estimator, while porosity-based predictions show the largest errors (6.36 for UCS, 2.20 for E). These results collectively demonstrate that non-destructive tests (especially $H_S$) and density measurements can reliably predict both strength and stiffness characteristics in geotechnical materials.

After developing the statistical models, their accuracy performances were evaluated using four well-known statistical indexes namely root mean square error (RMSE), mean absolute error (MAE), mean absolute percentage error (MAPE), and coefficient of determination ($R^2$) as follows [54]:

$$RMSE = \sqrt{\frac{1}{N} \sum_{i=1}^{N} (y - y')^2} \tag{16}$$

$$MAE = \left[ \frac{1}{N} \sum_{i=1}^{N} |y - \acute{y}| \right] \times 100 \tag{17}$$

$$MAPE = \left[ \frac{1}{N} \sum_{i=1}^{N} \left| \frac{y - y'}{y} \right| \right] \times 100 \tag{18}$$

$$R^2 = 1 - \frac{\sum_i (y_i - \acute{y}_i)^2}{\sum_i (y_i - \overline{y})^2} \tag{19}$$

**Table 8. Statistical parameters resulted from the simple regression analyses.**

| Model (y − x) | R | R² | Standard error | P(T<=t) two-tail | P-value |
|---|---|---|---|---|---|
| UCS − $\gamma_d$ | 0.95 | 0.90 | 5.16 | $5.55 \times 10^{-13}$ | $2.85 \times 10^{-10}$ |
| UCS − $G_s$ | 0.93 | 0.86 | 5.94 | $1.68 \times 10^{-15}$ | $3.36 \times 10^{-09}$ |
| UCS − $n_e$ | 0.92 | 0.84 | 6.36 | $1.11 \times 10^{-15}$ | $1.15 \times 10^{-08}$ |
| UCS − $H_S$ | 1.00 | 0.99 | 1.17 | $1.68 \times 10^{-11}$ | $6.54 \times 10^{-22}$ |
| UCS − PLI | 0.99 | 0.97 | 2.61 | $2.21 \times 10^{-15}$ | $1.21 \times 10^{-15}$ |
| UCS − BTS | 0.97 | 0.94 | 4.06 | $2.10 \times 10^{-15}$ | $3.36 \times 10^{-12}$ |
| E − $\gamma_d$ | 0.97 | 0.94 | 1.32 | $5.86 \times 10^{-15}$ | $1.30 \times 10^{-12}$ |
| E − $G_s$ | 0.95 | 0.90 | 1.74 | $9.96 \times 10^{-20}$ | $2.05 \times 10^{-10}$ |
| E − $n_e$ | 0.92 | 0.84 | 2.20 | $3.54 \times 10^{-24}$ | $1.39 \times 10^{-08}$ |
| E − $H_S$ | 0.98 | 0.97 | 1.02 | $2.59 \times 10^{-09}$ | $1.25 \times 10^{-14}$ |
| E − PLI | 0.98 | 0.95 | 1.19 | $3.40 \times 10^{-20}$ | $1.96 \times 10^{-13}$ |
| E − BTS | 0.95 | 0.91 | 1.64 | $2.97 \times 10^{-20}$ | $6.80 \times 10^{-11}$ |

Note: Perfect R = 1; Perfect R² = 1; Perfect standard error = 0; Perfect P(T<=t) two-tail = 1; Perfect P-value = 0

where y and y' are the experimental and predicted values of UCS and E, and N is the total number of data (20 rock samples). The values of the four above-mentioned indexes for the developed simple regression models are presented in Table 9. The predictive models demonstrate excellent performance in estimating unconfined compressive strength (UCS) and elasticity modulus (E), with particularly outstanding results for the UCS-$H_s$ model ($R^2 = 0.995$, RMSE = 1.271, MAPE = 0.2%), confirming the Schmidt hammer test as the most accurate non-destructive method for strength prediction. Dry unit weight-based models ($\gamma_d$) show strong performance for both UCS ($R^2 = 0.897$) and E ($R^2 = 0.943$), while effective porosity ($n_e$) models exhibit slightly lower but still robust predictive capability ($R^2 = 0.843-0.840$). All models maintain high accuracy, as evidenced by exceptionally low MAE (< 0.18) and MAPE (< 0.6%) values across all parameters. The point load index (PLI) demonstrates particularly reliable performance for both UCS ($R^2 = 0.974$) and E ($R^2 = 0.953$) estimation, while Brazilian tensile strength (BTS) shows slightly reduced but still strong predictive power ($R^2 = 0.936-0.911$). The remarkably low error metrics across all models validate their effectiveness for geotechnical property estimation, with mechanical test-based predictions ($H_s$, PLI, BTS) generally outperforming physical property-based models ($\gamma_d$, $G_s$, $n_e$).

## 4. Artificial intelligence-based analyses

In the field of computational intelligence, various algorithms with different approaches are developed recently. Three of these methods are used in this research namely Artificial Neural Network (ANN), K-Nearest Neighbors (kNN), and Random Forest (RF). The artificial intelligence–supported models employed in this study differ fundamentally in their learning mechanisms, data handling strategies, and prediction behavior, which justifies their comparative use. The Artificial Neural Network (ANN) is a parametric, nonlinear model capable of capturing complex interactions among input variables through hidden-layer neurons and weighted connections, making it particularly effective for modeling highly nonlinear relationships between physical–mechanical properties and UCS/E. In contrast, the K-Nearest Neighbors (kNN) algorithm is a non-parametric, instance-based learner that predicts outputs based on similarity measures in the feature space, making it sensitive to data distribution and scaling but advantageous for small datasets without requiring explicit training. The Random Forest (RF) model is an ensemble, tree-based method that combines multiple decision trees using bootstrap aggregation, improving generalization performance and reducing overfitting while providing insights into variable importance. By applying these conceptually distinct AI techniques to the same dataset, this study highlights their respective strengths and limitations in predicting sandstone UCS and elasticity modulus, with ANN demonstrating superior accuracy, kNN offering simplicity and transparency, and RF providing robustness and interpretability.

Table 9. Values of the performance indexes for evaluating the developed Simple Regression-based models.

| Model (y – x) | RMSE | MAE | MAPE | $R^2$ |
|---|---|---|---|---|
| UCS-$\gamma_d$ | 23.961 | 6.558 | 29.799 | 0.897 |
| UCS-$G_s$ | 31.791 | 0.178 | 0.597 | 0.863 |
| UCS-$n_e$ | 36.413 | 1.362 | 0.543 | 0.843 |
| UCS-$H_s$ | 1.271 | 17.819 | 0.211 | 0.995 |
| UCS-PLI | 6.151 | 0.393 | 0.097 | 0.974 |
| UCS-BTS | 14.82 | 1.082 | 0.225 | 0.936 |
| E-$\gamma_d$ | 1.566 | 5.210 | 0.167 | 0.943 |
| E-$G_s$ | 2.739 | 0.586 | 0.118 | 0.899 |
| E-$n_e$ | 4.359 | 0.315 | 0.162 | 0.840 |
| E-$H_s$ | 0.966 | 17.627 | 0.321 | 0.966 |
| E-PLI | 1.268 | 0.762 | 0.063 | 0.953 |
| E-BTS | 2.424 | 0.884 | 0.078 | 0.911 |

Note: Perfect $R^2 = 1$; Perfect RMSE = 0; Perfect MAE = 0%; Perfect MAPE = 0%; Perfect $R^2 = 1$

## 4.1. Artificial Neural Network (ANN)

The Artificial Neural Network (ANN) is a method used in classification and regression problems. When the output parameter in the data is qualitative and categorized, the problem is a classification type, and when it is numerical and continuous, it is a regression problem. The most important components of this method are processors called neurons. Their task is to discover intrinsic relationships between data by dividing the problem into three layers: input, output, and hidden. As shown in Fig 12, the hidden layer processes the information received from the input layer and passes it to the output layer. Each network learns by receiving training examples, ultimately leading to learning. During training, the weights of neurons in each layer are adjusted to minimize the error in estimating the output parameter. If the obtained error is within an acceptable range, the constructed model can be trusted [59].

## 4.2. K-Nearest Neighbors (kNN)

The K-Nearest Neighbors (kNN) method is a non-parametric approach used in classification and regression problems. In this method, the parameter K plays a key role. K represents the number of neighbors or the closest training samples to the target sample. In kNN regression, the output is the average of the K nearest neighbors. Since the learning phase of this method coincides with the testing phase and is performed in one step, it is also called a lazy algorithm. The selected samples can also be weighted based on their distance from the target sample. Additionally, this method uses various distance metrics, particularly Euclidean distance, to find neighboring samples. This distance is calculated using below equation:

$$d(X, Y) = \sqrt{\sum_{i=1}^{n} (x_i - y_i)^2}$$

(20)

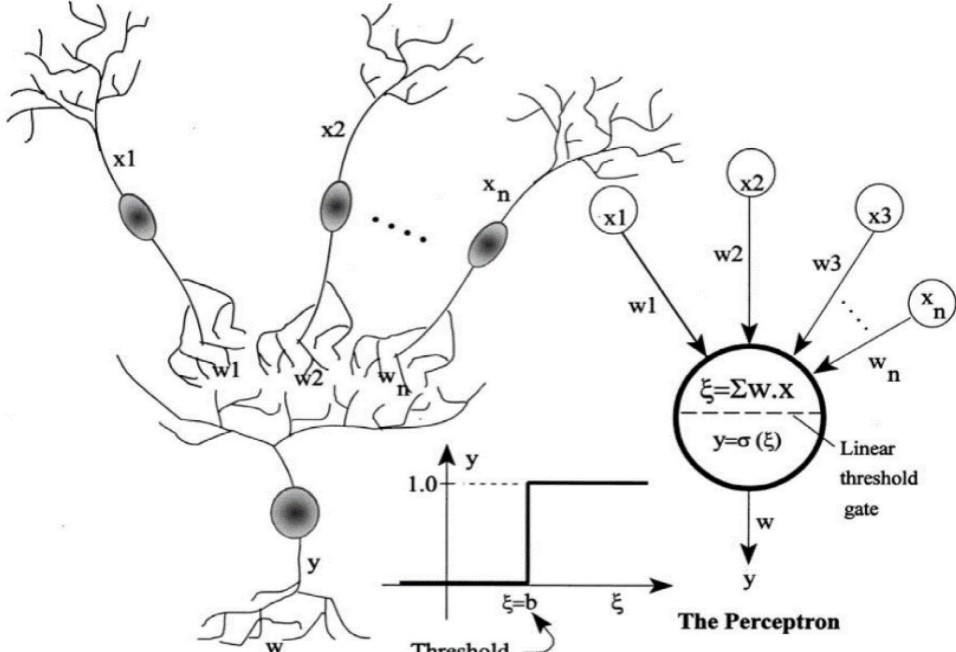

**Fig 12. Schematic overview of the artificial neural network's general operation [57,58].**

where x represents the training samples, y represents the test samples, and i is the number of parameters. The training samples are then sorted in ascending order based on their distance, and the top K samples are selected for estimating the target variable. Determining the number of neighbors (K) is one of the most critical steps in this method, which is done through trial and error [60].

### 4.3. Random Forest (RF)

The Random Forest (RF) method is a tree-based approach used for regression and classification problems. It is a relatively complex method that aims to improve model accuracy by training multiple decision trees and combining them. Each decision tree is trained with a sample, and the selection of predictor variables used for splitting nodes is random. In Random Forest, two parameters, $m_{try}$ and $n_{tree}$, play essential roles and must be properly assigned. $m_{try}$ is the number of auxiliary variables used in each decision tree and can range from one to the total number of auxiliary variables. $n_{tree}$ is the total number of decision trees, typically set between 500 and 1000 by the user. Fig 13 shows a Random Forest consisting of three or more decision trees [61].

The next objective is to apply several computational intelligence algorithms to predict the parameters UCS and E. For this purpose, using the Orange 3.39.0 software, models of Artificial Neural Network (ANN), K-Nearest Neighbors (kNN), and Random Forest (RF) were developed. Fig 14 shows the Orange 3.39.0 software workspace with the required operators for modeling, including File (for Adding dataset or data), Data Table (for seeing the data) Preprocess (for initial data processing), Data Sampler (for dividing the data into training and test sets), Select Columns (for selecting independent and dependent variables), Test and Score (for model evaluation), and Rank (for parameter ranking).

The evaluation and validation results of the three artificial intelligence-based techniques are presented in Tables 10–12 and depicted in Fig 15. For comparing the performance of the methods, four types of errors were used (RMSE, MAE, MAPE, and $R^2$). The RMSE is a measure of prediction error relative to the actual value, where a lower value indicates higher accuracy. It is derived by taking the square root of MSE (Mean Squared Error). Unlike MSE, RMSE has the same unit as the target parameter. MAE, which uses absolute errors instead of squared errors, provides a value with the same unit but is non-differentiable and can pose challenges in gradient-based optimization.

Based on the obtained results, the NN models generally exhibit strong performance, particularly for UCS-H$_s$ (RMSE: 1.562, MAE: 0.977, MAPE: 1.081%, $R^2$: 0.992), indicating near-perfect prediction accuracy. The E-H$_s$ model also performs well ($R^2$: 0.947). However, UCS-n$_e$ has the weakest performance (RMSE: 6.96, $R^2$: 0.835), suggesting difficulty in predicting this parameter. The E-n$_e$ model also struggles ($R^2$: 0.718), implying that neural networks may not generalize well for certain rock properties. The kNN models show higher errors and lower $R^2$ values compared to NN models, indicating

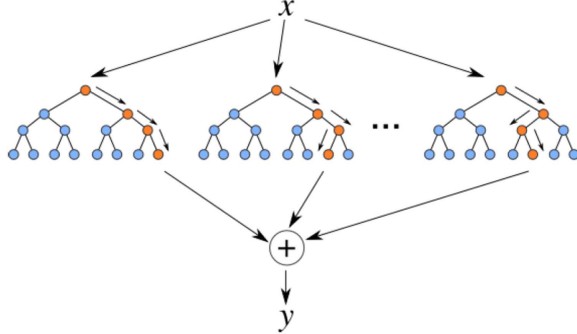

**Fig 13. Input of a test sample (x) into the Random Forest and output of the result (y).**

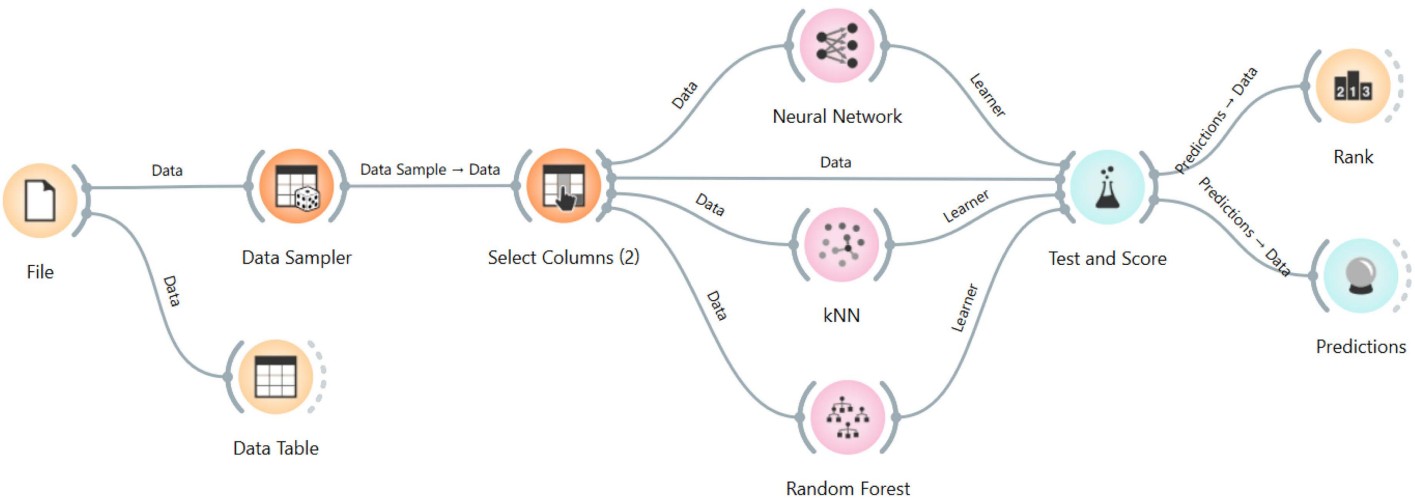

**Fig 14. Schematic of the Orange 3.39.0 software workspace after modeling.**

**Table 10. Values of the performance indexes for evaluating the developed ANN-based models.**

| Model | RMSE | MAE | MAPE | $R^2$ |
|---|---|---|---|---|
| UCS-γd | 4.433 | 3.125 | 3.564 | 0.933 |
| UCS-Gs | 5.191 | 3.322 | 3.965 | 0.908 |
| UCS-ne | 6.960 | 4.939 | 5.612 | 0.835 |
| UCS-Hs | 1.562 | 0.977 | 1.081 | 0.992 |
| UCS-PLI | 3.201 | 2.586 | 3.229 | 0.965 |
| UCS-BTS | 5.435 | 4.032 | 5.304 | 0.899 |
| E-γd | 1.491 | 1.163 | 2.365 | 0.934 |
| E-Gs | 1.974 | 1.625 | 3.391 | 0.884 |
| E-ne | 3.081 | 2.371 | 4.676 | 0.718 |
| E-Hs | 1.335 | 1.207 | 2.499 | 0.947 |
| E-PLI | 1.509 | 1.234 | 2.561 | 0.932 |
| E-BTS | 1.810 | 1.521 | 3.193 | 0.903 |

inferior predictive performance. The best-performing kNN model is UCS-$H_s$ (RMSE: 5.243, $R^2$: 0.906), but it still under-performs the NN equivalent. The E-$H_s$ model ($R^2$: 0.854) is the best among the "E" category but remains weaker than NN. The E-$G_s$ and E-$n_e$ models perform poorly ($R^2$: 0.793 and 0.780, respectively), reinforcing that kNN struggles with certain geotechnical predictions. The RF models show mixed performance, with some cases outperforming kNN but rarely matching NN. The UCS-PLI model excels (RMSE: 5.178, $R^2$: 0.909), while UCS-$G_s$ performs worst ($R^2$: 0.769). In the "E" category, E-$H_s$ stands out (RMSE: 1.763, $R^2$: 0.908), nearly matching NN performance, but other models (e.g., E-$n_e$, $R^2$: 0.759) are weaker.

As general findings, the Neural Networks are the best choice for UCS/E prediction, offering the highest $R^2$, lowest errors, and best generalizability. The Random Forest is competitive for specific cases (e.g., PLI-UCS, $H_s$-E) but struggles with density/porosity inputs. The kNN is the least accurate, with higher errors and lower $R^2$, but may suffice for quick, approximate estimates. The Best predictors are Hs > PLI > $\gamma_d$ > $G_s$ > $n_e$. $H_s$ and PLI should be prioritized for field testing when possible. For high-stakes applications (e.g., structural design), NN models with $H_s$/PLI inputs are ideal. For rapid

**Table 11. Values of the performance indexes for evaluating the developed kNN-based models.**

| Model | RMSE | MAE | MAPE | R2 |
|---|---|---|---|---|
| UCS-γd | 6.273 | 5.214 | 6.839 | 0.866 |
| UCS-Gs | 7.561 | 6.006 | 7.653 | 0.805 |
| UCS-ne | 6.754 | 5.624 | 7.298 | 0.845 |
| UCS-Hs | 5.243 | 4.182 | 5.816 | 0.906 |
| UCS-PLI | 5.928 | 4.797 | 6.421 | 0.880 |
| UCS-BTS | 6.821 | 5.598 | 7.276 | 0.842 |
| E-γd | 2.166 | 1.816 | 3.806 | 0.861 |
| E-Gs | 2.640 | 2.024 | 4.226 | 0.793 |
| E-ne | 2.721 | 2.319 | 4.771 | 0.780 |
| E-Hs | 2.218 | 1.953 | 4.101 | 0.854 |
| E-PLI | 2.269 | 1.932 | 4.047 | 0.847 |
| E-BTS | 2.565 | 2.231 | 4.603 | 0.805 |

**Table 12. Values of the performance indexes for evaluating the developed RF-based models.**

| Model | RMSE | MAE | MAPE | R2 |
|---|---|---|---|---|
| UCS-γd | 6.101 | 5.250 | 6.741 | 0.873 |
| UCS-Gs | 8.231 | 5.813 | 7.077 | 0.769 |
| UCS-ne | 6.768 | 5.482 | 6.745 | 0.844 |
| UCS-Hs | 5.884 | 4.745 | 6.562 | 0.882 |
| UCS-PLI | 5.178 | 4.269 | 5.539 | 0.909 |
| UCS-BTS | 6.318 | 5.243 | 6.765 | 0.864 |
| E-γd | 2.724 | 2.380 | 5.026 | 0.780 |
| E-Gs | 2.752 | 2.111 | 4.400 | 0.775 |
| E-ne | 2.850 | 2.409 | 4.848 | 0.759 |
| E-Hs | 1.763 | 1.464 | 2.998 | 0.908 |
| E-PLI | 2.182 | 1.977 | 4.095 | 0.859 |
| E-BTS | 2.346 | 2.005 | 4.127 | 0.837 |

field estimates, RF (PLI-based) or even kNN may be acceptable. Dry unit weight ($\gamma_d$) is a reliable fallback when mechanical test data is unavailable.

## 4.4. Sensitivity analysis

Sensitivity analysis involves understanding the influence of independent variables on the dependent variable and examining their relative importance within a dataset. Various methods for sensitivity analysis have been developed for classification and regression problems. Below, two methods used for ranking input parameters are discussed.

**4.4.1. Univariate regression method.** The Univariate Regression method, or single-variable feature selection, examines each feature individually to determine its relationship strength with the response variable. This method is simple to implement and understand and is generally suitable for better data comprehension. In classification problems, features are evaluated by comparing their values across the response variable's classes. Statistical tests such as t-tests, ANOVA, and others are used to determine whether the examined feature differs significantly across classes [62].

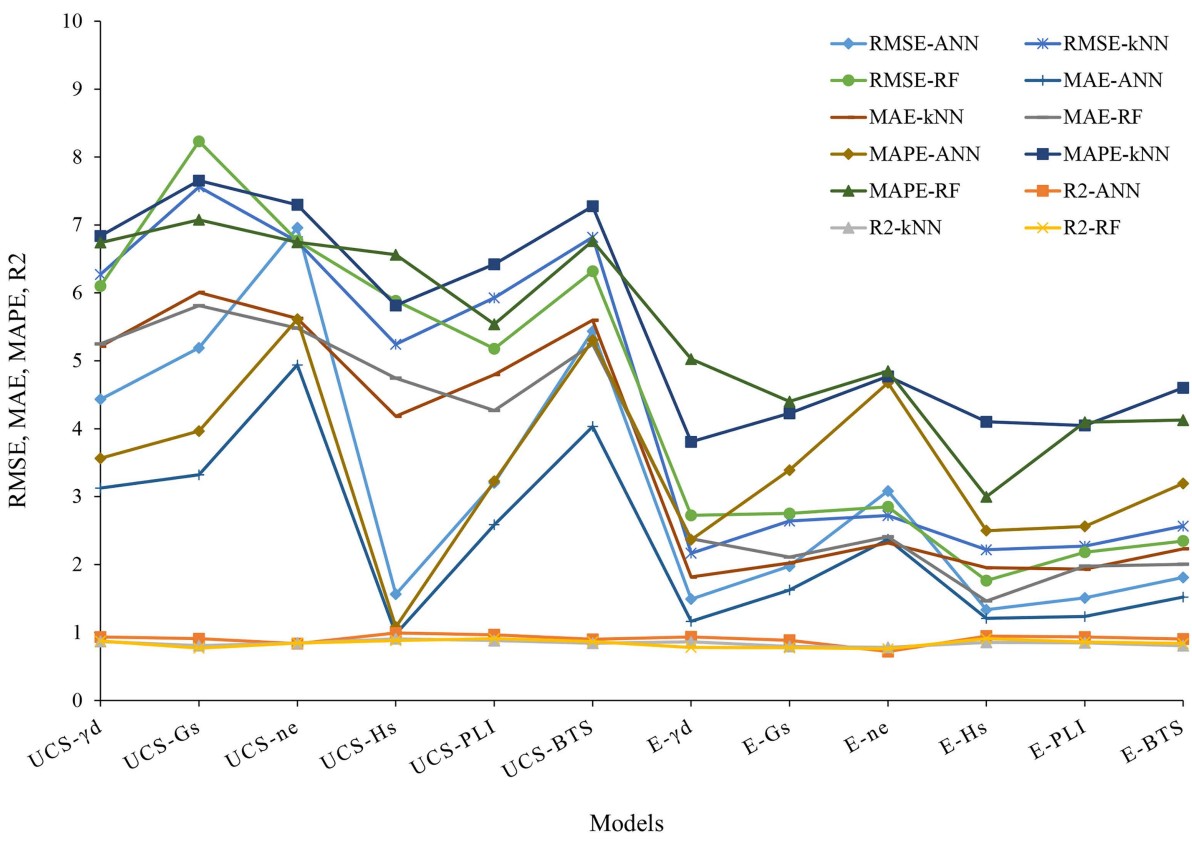

**Fig 15. Graphs of the performance indexes of the artificial intelligence-based models.**

**4.4.2. ReliefF method.** This method includes an algorithm developed by Kira and Rendell [63]. It uses a filter-based approach for feature selection and is highly sensitive to the relationships between features or parameters. Initially designed for binary classification problems, it was later extended to various prediction tasks. In this method, a score is calculated for each feature, and features are ranked and selected based on this score. Scoring is done by identifying differences in feature values between neighboring sample pairs. If the difference in a feature value is observed in a neighboring pair with the same class or output parameter value, the feature score decreases [64].

In modeling and incorporating different input parameters, sensitivity analysis of the output parameter relative to the inputs is crucial for understanding their influence. Here, two methods, Univariate Regression and RReliefF, were used to rank the importance of parameters. Table 13 presents the scores assigned to each parameter based on these two criteria, ranked by their importance. Based on the results, UCS-$H_s$ emerges as the most significant parameter, achieving the highest Univariate Regression score (3406.678) and a strong ReliefF value (0.633), securing the top rank. Following closely is UCS-PLI (Rank 2), which shows a notable difference between its Univariate Regression score (574.573) and ReliefF score (0.643), suggesting it is highly influential in predictive modeling. The E-$H_s$ and E-PLI parameters rank 3rd and 4th, respectively, maintaining relatively high scores in both methods, indicating their consistent importance. Mid-tier parameters like E-$\gamma_d$ (Rank 5) and E-$G_s$ (Rank 6) show moderate importance, while UCS-BTS (Rank 7) has a lower Univariate Regression score (196.144) but a surprisingly high ReliefF score (0.649), hinting at its contextual relevance. The lower-ranked parameters, such as UCS-$G_s$ (Rank 10), UCS-$n_e$ (Rank 11), and E-$n_e$ (Rank 12), exhibit the weakest scores, suggesting they contribute the least to model performance. Interestingly, while the Univariate Regression scores

**Table 13. Ranking and scoring of input parameters.**

| Model | Univariate regression method | ReliefF method | Rank |
|---|---|---|---|
| UCS-$H_s$ | 3406.678 | 0.633 | 1 |
| UCS-PLI | 574.573 | 0.643 | 2 |
| E-$H_s$ | 428.406 | 0.616 | 3 |
| E-PLI | 346.311 | 0.630 | 4 |
| E-$\gamma_d$ | 317.026 | 0.629 | 5 |
| E-$G_s$ | 201.256 | 0.588 | 6 |
| UCS-BTS | 196.144 | 0.649 | 7 |
| UCS-$\gamma_d$ | 184.861 | 0.634 | 8 |
| E-BTS | 144.686 | 0.634 | 9 |
| UCS-$G_s$ | 127.183 | 0.589 | 10 |
| UCS-$n_e$ | 102.028 | 0.607 | 11 |
| E-$n_e$ | 84.207 | 0.589 | 12 |

vary dramatically (from 3406.678 to 84.207), the ReliefF scores remain in a tighter range (0.588–0.649), indicating that the latter method provides more balanced feature weighting. Overall, the ranking highlights UCS-$H_s$ and UCS-PLI as the most critical predictors, while parameters like UCS-$n_e$ and E-$n_e$ are the least influential, which could guide feature selection in geotechnical machine learning models.

## 5. Conclusions

This research developed a robust framework for predicting the uniaxial compressive strength (UCS) and elasticity modulus (E) of sandstones by combining detailed laboratory testing with statistical analysis and machine learning algorithms. The study analyzed 20 samples from four Iranian formations, revealing that geological context is a primary determinant of mechanical behavior. The PDH formation, characterized by high density ($\gamma_d$ = 25.52–25.99 kN/m³) and low porosity ($n_e$ = 2.48–4.84%), exhibited the highest strength (UCS = 97.21 MPa, E = 54.82 GPa). In contrast, the more porous HZD formation ($n_e$ = 8.02–10.16%) displayed significantly weaker properties (UCS = 60.25 MPa, E = 41.87 GPa), underscoring the necessity of formation-specific models in geotechnical practice.

Strong quantitative relationships were established between physical and mechanical properties. Dry unit weight ($\gamma_d$) showed the strongest positive correlations with UCS (R = 0.947) and E (R = 0.971), while porosity ($n_e$) was strongly negatively correlated (R = −0.918 and −0.916, respectively). These results confirm density as a key indicator of strength and porosity as a critical weakening factor.

Among the artificial intelligence models evaluated, Artificial Neural Networks (ANN) achieved the highest predictive accuracy, particularly when using Schmidt hammer rebound ($H_s$) as input (UCS-$H_s$: $R^2$ = 0.992; E-$H_s$: $R^2$ = 0.947). ANN consistently outperformed Random Forest and k-Nearest Neighbors models, demonstrating its superior ability to capture the complex, non-linear relationships inherent in the data. This comparison provides clear guidance for selecting modeling approaches based on the required level of precision in practical applications.

The practical significance of this research lies in the validated integration of formation-specific laboratory data with comparative statistical and artificial intelligence–based modeling for predicting sandstone UCS and elasticity modulus. By systematically evaluating conventional regression approaches alongside ANN, kNN, and RF models using identical input parameters, the study demonstrates the clear performance advantage of ANN for site-specific applications. The results provide a reliable framework for preliminary geotechnical assessment in sandstone formations where direct testing is limited, while the formation-based interpretation and sensitivity analysis enhance confidence in model applicability. Overall, this work contributes to rock mechanics by offering a structured, data-driven methodology that improves

prediction accuracy and supports informed engineering decision-making rather than by reiterating established empirical relationships.

## Author contributions

**Conceptualization:** Davood Fereidooni.

**Formal analysis:** Matloob Hejazifar.

**Funding acquisition:** Davood Fereidooni.

**Investigation:** Davood Fereidooni.

**Methodology:** Davood Fereidooni.

**Project administration:** Davood Fereidooni.

**Software:** Davood Fereidooni, Matloob Hejazifar.

**Supervision:** Davood Fereidooni.

**Validation:** Davood Fereidooni.

**Visualization:** Davood Fereidooni.

**Writing – original draft:** Davood Fereidooni, Matloob Hejazifar.

**Writing – review & editing:** Davood Fereidooni.

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
