## [Decision Letter · Decision Letter 0]

14 Dec 2025

PONE-D-25-54958

Predicting the uniaxial compressive strength and elasticity modulus of sandstones from physical and mechanical properties using statistical analyses and artificial intelligence-based techniques

PLOS One

Dear Dr. Fereidooni,

Thank you for submitting your manuscript to PLOS ONE. After careful consideration, we feel that it has merit but does not fully meet PLOS ONE’s publication criteria as it currently stands. Therefore, we invite you to submit a revised version of the manuscript that addresses the points raised during the review process.

If applicable, we recommend that you deposit your laboratory protocols in protocols.io to enhance the reproducibility of your results. Protocols.io assigns your protocol its own identifier (DOI) so that it can be cited independently in the future. For instructions see: https://journals.plos.org/plosone/s/submission-guidelines#loc-laboratory-protocols. Additionally, PLOS ONE offers an option for publishing peer-reviewed Lab Protocol articles, which describe protocols hosted on protocols.io. Read more information on sharing protocols at . Additionally, PLOS ONE offers an option for publishing peer-reviewed Lab Protocol articles, which describe protocols hosted on protocols.io. Read more information on sharing protocols at https://plos.org/protocols?utm_medium=editorial-email&utm_source=authorletters&utm_campaign=protocols..

We look forward to receiving your revised manuscript.

Kind regards,

Shamshad Alam, PhD

Academic Editor

PLOS One

Journal Requirements:

“This work is based upon research funded by Iran National Science Foundation (INSF) under project No. 4040158.”

“This work is based upon research funded by Iran National Science Foundation (INSF) under project No. 4040158.”

6. In the online submission form you indicate that your data is not available for proprietary reasons and have provided a contact point for accessing this data. Please note that your current contact point is a co-author on this manuscript. According to our Data Policy, the contact point must not be an author on the manuscript and must be an institutional contact, ideally not an individual. Please revise your data statement to a non-author institutional point of contact, such as a data access or ethics committee, and send this to us via return email. Please also include contact information for the third party organization, and please include the full citation of where the data can be found.

8. We note that Figure 2 in your submission contain map/satellite images which may be copyrighted. All PLOS content is published under the Creative Commons Attribution License (CC BY 4.0), which means that the manuscript, images, and Supporting Information files will be freely available online, and any third party is permitted to access, download, copy, distribute, and use these materials in any way, even commercially, with proper attribution. For these reasons, we cannot publish previously copyrighted maps or satellite images created using proprietary data, such as Google software (Google Maps, Street View, and Earth). For more information, see our copyright guidelines: http://journals.plos.org/plosone/s/licenses-and-copyright.

1. You may seek permission from the original copyright holder of Figure 2 to publish the content specifically under the CC BY 4.0 license.

Reviewers' comments:

Reviewer's Responses to Questions

**Comments to the Author**

1. Is the manuscript technically sound, and do the data support the conclusions?

Reviewer #1: Yes

Reviewer #2: Partly

2. Has the statistical analysis been performed appropriately and rigorously? 

Reviewer #1: Yes

Reviewer #2: Yes

3. Have the authors made all data underlying the findings in their manuscript fully available?

Reviewer #1: Yes

Reviewer #2: Yes

4. Is the manuscript presented in an intelligible fashion and written in standard English?

Reviewer #1: Yes

Reviewer #2: Yes

5. Review Comments to the Author

Reviewer #1: My comments on the manuscript titled "Predicting the uniaxial compressive strength and elasticity modulus of sandstones from physical and mechanical properties using statistical analyses and artificial intelligence-based techniques" are below;

- The novelty of the article is not stated in the Introduction. Numerous studies exist on predicting the UCS and elasticity of different rock types based on the properties of other rocks. Furthermore, many studies have used artificial intelligence. Therefore, the authors should mention the novelty of their work.

- The authors noted that the samples were washed and dried before laboratory testing. It should be noted that this could alter the mechanical properties of the samples, particularly UCS. A statement regarding this should be added to the relevant section.

- The authors mentioned that "The petrographic investigations were done by optical microscopy .."in Section 2.2. However, no results were given regarding the mineralogical properties of the rocks, and they were not used in the manuscript.

- Table 1 shows that the physical and mechanical properties of samples selected from the same region vary within a narrow range. The implications of this for statistical analyses should be discussed. Furthermore, the suitability of the resulting equations for general use should also be evaluated.

- It is already known that there are significant relationships between UCS values of rocks and, in particular, point load and Schmidt hardness values. Indeed, models have been developed to estimate UCS from point load and Schmidt hardness values. Therefore, it is expected that the models obtained in this study would also have high correlation values. This limits the originality of the study.

- The differences between the artificial intelligence-supported models used should be given in detail.

- Some explanations such as "The demonstrated effectiveness of non-destructive methods, particularly Schmidt hammer and point load tests, provides viable alternatives to conventional laboratory testing, especially in challenging field conditions." given in the Conclusion sectionare repetitive of results found in many previous studies. Therefore, it is understood that the study does not contain original results.

Reviewer #2: The following comments need to be addressed for further improvement of paper.

1. This paper explored the application of the AI techniques for the prediction of UCS and Elastic modulus based on the laboratory tests data. The findings of the paper

may be helpful the rock engineering community. However, during the review it was seen that the abstract doesn't contain the performance of the used AI models.

2. The introduction section need to be revised by including the latest literature. In this regard some of the literature is given as: Multiple stress memory characteristic of rocks under uniaxial deflection loading: Insights from acoustic emission signals, Compressive damage constitutive model for brittle coal based on the compaction effect and linear energy dissipation law , Fracture features of brittle coal under uniaxial and cyclic compression loads, Numerical

analysis on the factors affecting post-peak characteristics of coal under uniaxial compression, Localization of Acoustic Emission Source in Rock Using SMIGWO Algorithm, Study on seismic displacement of coal-rock fracture based on radiation energy, Mechanism and research progress of ultrasonic

excitation flow enhancement in low permeability coal seam, Slope stability prediction based on a long short-term memory neural network: comparisons with convolutional neural networks, support vector machines and random forest models, Uncertainties of landslide susceptibility prediction: influences of different study area scales and mapping unit scales, Appraisal of Different Artificial Intelligence Techniques for the Prediction of Marble Strength, An Appropriate Model for the Prediction of Rock Mass Deformation Modulus among Various Artificial Intelligence Models, Application of Machine Learning and Multivariate Statistics to Predict Uniaxial Compressive Strength and Static Young’s Modulus Using Physical Properties under Different Thermal Conditions.

3. Authors are advised to include the correlation matrices analysis of the input variable used for the prediction of UCS.

4. Adiscussion section need to be included and the findings of the research need to be discussed.

5. The conclusion section need to be summarized and the main findings shall be included

6. PLOS authors have the option to publish the peer review history of their article (what does this mean?). If published, this will include your full peer review and any attached files.). If published, this will include your full peer review and any attached files.

.

Reviewer #1: No

Reviewer #2: **Yes:**Sajjad HussainSajjad Hussain

---

## [Author Response · Author response to Decision Letter 1]

27 Jan 2026

REVISION NOTES

Manuscript number: PONE-D-25-54958

Title:

“Predicting the uniaxial compressive strength and elasticity modulus of sandstones from physical and mechanical properties using statistical analyses and artificial intelligence-based techniques”

December 23, 2025

Dear Editor,

I would like to sincerely thank you and the reviewers for your valuable and constructive comments and suggestions on our manuscript. After carefully reviewing all the feedback, we have revised the manuscript accordingly and have tried to address all recommendations as thoroughly as possible. The detailed responses to each of the reviewers’ comments are listed below, point by point. The changes made to the manuscript are highlighted in red in the file titled "Manuscript - Marked-up Version (R1)." It should be noted that all the journal requirements were considered and applied in the text of the manuscript and during its online submission to the journal. This work is based upon research funded by Iran National Science Foundation (INSF) under project No. 4040158. The funder had no role in study design, data collection and analysis, decision to publish, or preparation of the manuscript. We respectfully submit the revised version of the manuscript for your kind consideration and would be grateful if you would evaluate the possibility of its publication. Should you require any further clarifications or additional information, please do not hesitate to contact me. I look forward to hearing from you soon.

Sincerely,

Corresponding author

Journal Requirements:

Reply; The comment was considered.

Reply; The comment was considered. The text was modified based on this comment. Please see the first paragraph of the “Research steps and materials”.

Reply; The comment was considered. The “Declarations” part was removed from the text.

“This work is based upon research funded by Iran National Science Foundation (INSF) under project No. 4040158.”

Reply; The comment was considered. The required statements were mentioned in the cover letter.

“This work is based upon research funded by Iran National Science Foundation (INSF) under project No. 4040158.”

Reply; The comment was considered.

6. In the online submission form you indicate that your data is not available for proprietary reasons and have provided a contact point for accessing this data. Please note that your current contact point is a co-author on this manuscript. According to our Data Policy, the contact point must not be an author on the manuscript and must be an institutional contact, ideally not an individual. Please revise your data statement to a non-author institutional point of contact, such as a data access or ethics committee, and send this to us via return email. Please also include contact information for the third party organization, and please include the full citation of where the data can be found.

Reply; The comment was considered.

Reply; The comment was considered.

8. We note that Figure 2 in your submission contain map/satellite images which may be copyrighted. All PLOS content is published under the Creative Commons Attribution License (CC BY 4.0), which means that the manuscript, images, and Supporting Information files will be freely available online, and any third party is permitted to access, download, copy, distribute, and use these materials in any way, even commercially, with proper attribution. For these reasons, we cannot publish previously copyrighted maps or satellite images created using proprietary data, such as Google software (Google Maps, Street View, and Earth). For more information, see our copyright guidelines:

http://journals.plos.org/plosone/s/licenses-and-copyright.

Reply: The authors of the manuscript have drawn a general map of Iran by themselves and provided required information on the map that it doesn’t need to any license or copyright permission.

1. You may seek permission from the original copyright holder of Figure 2 to publish the content specifically under the CC BY 4.0 license.

Reply; The comment was considered.

Response to the comments of Reviewer #1

We appreciate the reviewer’s constructive feedback. We have substantially revised the manuscript to (1) clarify the analytical comparisons among classification systems, (2) strengthen the methodological justification and parameter derivations, (3) improve figures, tables, and writing clarity, and (4) highlight the broader scientific and international relevance of the study.

Reviewer #1: My comments on the manuscript titled "Predicting the uniaxial compressive strength and elasticity modulus of sandstones from physical and mechanical properties using statistical analyses and artificial intelligence-based techniques" are below;

- The novelty of the article is not stated in the Introduction. Numerous studies exist on predicting the UCS and elasticity of different rock types based on the properties of other rocks. Furthermore, many studies have used artificial intelligence. Therefore, the authors should mention the novelty of their work.

Reply; The comment is accepted. We added a new descriptive paragraph for providing the novelty of the work to the Introduction part of the manuscript as: … “The novelty of this study lies in its integrated and systematic framework that combines comprehensive statistical analyses (covariance determination, multiple correlation metrics, and simple regression) with a comparative evaluation of multiple artificial intelligence techniques (ANN, kNN, and RF) specifically applied to sandstones from four distinct Iranian geological formations. Unlike many previous studies that focus on either statistical approaches or a single AI method, this research simultaneously evaluates traditional and AI-based models using an identical dataset, allowing for a transparent performance comparison and robustness assessment. In addition, the study emphasizes the use of easily obtainable, non-destructive physical and mechanical parameters and provides detailed sensitivity analysis to quantify the relative importance of input variables, identifying Schmidt hammer rebound as the most influential predictor for both UCS and elasticity modulus. The formation-based interpretation of results further distinguishes this work by linking predictive performance to petro-physical variability among sandstones, thereby enhancing the practical applicability of the proposed models for site-specific geotechnical design where direct laboratory testing is limited.”

- The authors noted that the samples were washed and dried before laboratory testing. It should be noted that this could alter the mechanical properties of the samples, particularly UCS. A statement regarding this should be added to the relevant section.

Reply; We appreciate this insightful comment. In this study, washing and oven-drying the specimens at 105 °C were performed solely to remove surface dust, cutting residues, and free moisture, in accordance with ISRM (2007) recommendations, and to ensure consistent initial conditions for physical property measurements. This procedure is widely adopted in rock mechanics studies and is not expected to significantly alter the intrinsic mechanical behavior of intact sandstones, particularly for dense, low- to moderately porous materials such as those investigated here. Also, these specimens experienced moisture and water absorption once during coring with a coring machine and cutting off their heads and bottoms, and the washing operation was performed after cutting off the heads and bottoms of the specimens with a cutter.

So, all specimens were subjected to the specimen preparation protocol, ensuring that any potential minor influence of drying on UCS and elasticity modulus would be systematic and would not affect the comparative analysis, correlations, or predictive modeling results.

- The authors mentioned that "The petrographic investigations were done by optical microscopy .."in Section 2.2. However, no results were given regarding the mineralogical properties of the rocks, and they were not used in the manuscript.

Reply; We thank the reviewer for this important observation. In this study, the petrographic investigations were conducted to ensure proper lithological identification, and to identify correct lithological name of the studied sandstones. The related text was corrected based on this comment.

- Table 1 shows that the physical and mechanical properties of samples selected from the same region vary within a narrow range. The implications of this for statistical analyses should be discussed. Furthermore, the suitability of the resulting equations for general use should also be evaluated.

Reply; We appreciate this constructive comment. The relatively narrow range of physical and mechanical properties observed within samples from the same formation reflects their similar depositional environment, mineralogical composition, and diagenetic history, which is typical for sandstones originating from a single geological unit. From a statistical perspective, this homogeneity enhances internal consistency and reduces noise, allowing more reliable identification of intrinsic relationships between input parameters and UCS and elasticity modulus, as evidenced by the high correlation coefficients and low prediction errors. However, it is acknowledged that limited intra-formation variability may constrain the extrapolation of the derived regression equations beyond the studied property ranges. Accordingly, the developed equations and AI models are most suitable for sandstones with comparable lithological characteristics and physical property ranges, and their direct application to other rock types or highly heterogeneous sandstones should be undertaken with caution. This limitation was discussed in the manuscript.

- It is already known that there are significant relationships between UCS values of rocks and, in particular, point load and Schmidt hardness values. Indeed, models have been developed to estimate UCS from point load and Schmidt hardness values. Therefore, it is expected that the models obtained in this study would also have high correlation values. This limits the originality of the study.

Reply; We acknowledge the reviewer’s point and agree that strong correlations between UCS, point load index, and Schmidt rebound hardness are well established in the literature. In my opinion, the correlations between UCS, point load index, and Schmidt rebound hardness are not very bad based on the obtained coefficient of determination (R2) and other indicators (please see Tables 8, 9, 10, 11 and 12). However, the originality of this study does not stem from the mere existence of these correlations, but from the way they are systematically quantified, compared, and integrated within a unified analytical framework. Unlike many previous studies that focus on developing a single empirical equation, this research concurrently evaluates classical statistical models and multiple AI-based techniques using the same dataset, allowing a transparent assessment of their relative predictive perf

---

## [Decision Letter · Decision Letter 1]

30 Mar 2026

Predicting the uniaxial compressive strength and elasticity modulus of sandstones from physical and mechanical properties using statistical analyses and artificial intelligence-based techniques

PONE-D-25-54958R1

Dear Dr. Fereidooni,

We’re pleased to inform you that your manuscript has been judged scientifically suitable for publication and will be formally accepted for publication once it meets all outstanding technical requirements.

An invoice will be generated when your article is formally accepted. Please note, if your institution has a publishing partnership with PLOS and your article meets the relevant criteria, all or part of your publication costs will be covered. Please make sure your user information is up-to-date by logging into Editorial Manager at Editorial Manager® and clicking the ‘Update My Information' link at the top of the page. For questions related to billing, please contact  and clicking the ‘Update My Information' link at the top of the page. For questions related to billing, please contact billing support..

Kind regards,

Shamshad Alam, PhD

Academic Editor

PLOS One

Additional Editor Comments (optional):

Reviewers' comments:

Reviewer's Responses to Questions

**Comments to the Author**

1. If the authors have adequately addressed your comments raised in a previous round of review and you feel that this manuscript is now acceptable for publication, you may indicate that here to bypass the “Comments to the Author” section, enter your conflict of interest statement in the “Confidential to Editor” section, and submit your "Accept" recommendation.

Reviewer #3: All comments have been addressed

Reviewer #4: All comments have been addressed

2. Is the manuscript technically sound, and do the data support the conclusions?

Reviewer #3: Yes

Reviewer #4: Yes

3. Has the statistical analysis been performed appropriately and rigorously? 

Reviewer #3: Yes

Reviewer #4: Yes

4. Have the authors made all data underlying the findings in their manuscript fully available?

Reviewer #3: Yes

Reviewer #4: Yes

5. Is the manuscript presented in an intelligible fashion and written in standard English?

Reviewer #3: Yes

Reviewer #4: Yes

6. Review Comments to the Author

Reviewer #3: The objectives of the work in this manuscript are relatively clear, the methods are standardized, the results are reasonable, the research process is introduced in detail, and the questions raised by the previous reviewers have been answered and revised.

Reviewer #4: The authors have fully addressed all the questions and concerns raised by the reviewers, providing clear and reasonable explanations for each point. I recommend acceptance of the manuscript as it meets the academic standards and contributes valuable insights to the field.

7. PLOS authors have the option to publish the peer review history of their article (what does this mean?). If published, this will include your full peer review and any attached files.). If published, this will include your full peer review and any attached files.

.

Reviewer #3: No

Reviewer #4: No

---

## [Editor Report · Acceptance letter]

PONE-D-25-54958R1

PLOS One

Dear Dr. Fereidooni,

I'm pleased to inform you that your manuscript has been deemed suitable for publication in PLOS One. Congratulations! Your manuscript is now being handed over to our production team.

Kind regards,

on behalf of

Dr. Shamshad Alam

Academic Editor

PLOS One